# Introducing Electronic Circuits and Hydrological Models to Postsecondary Physical Geography and Environmental Science Students: Systems Science, Circuit Theory, Construction and Calibration

Nicholas J. Kinar[1,2,3,4]
[1]Smart Water Systems Lab, University of Saskatchewan
[2]Global Institute for Water Security, University of Saskatchewan
[3]Centre for Hydrology, University of Saskatchewan
[4]Department of Geography and Planning, University of Saskatchewan

*Correspondence to*: Nicholas J. Kinar (n.kinar@usask.ca)

**Abstract.** A classroom activity involving the construction, calibration and testing of electronic circuits was introduced to an advanced hydrology class at the postsecondary level. Two circuits were constructed by students: (1) a water detection circuit; and (2) a hybrid relative humidity (RH)/air temperature sensor and pyranometer. The circuits motivated concepts of systems science, modelling in hydrology, and model calibration. Students used the circuits to collect data useful for

providing inputs to mathematical models of hydrological processes. Each student was given the opportunity to create a custom hydrological model within the context of the class. This is an example of constructivist teaching where students engage in the creation of meaningful knowledge and the instructor serves as a facilitator to assist students in the achievement of a goal. Analysis of student-provided feedback showed that the circuit activity motivated, engaged, and facilitated learning. Students also found the activity to be a novel and enjoyable experience. The theory of circuit operation and

calibration is provided along with a complete bill of materials (BOM) and design files for replication of this activity in other postsecondary classrooms. Student suggestions for improvement of the circuit activity are presented along with additional applications.

## 1 Introduction

Due to the increasing need for interdisciplinary approaches in hydrology (Vogel et al., 2015), teaching of this
subject at the post-secondary level should utilize a synthesis of techniques that involve the introduction of concepts and theories with an emphasis on real-world applications (Seibert et al., 2013; Van Loon, 2019). To maximize societal gain, these applications can address United Nations Sustainable Development Goals (SDGs) to encourage environmental stewardship, human equality, and a basic standard of living while preserving the functionality of planetary systems that sustain life (Crespo et al., 2017; Filho et al., 2019; Kopnina, 2018). Closely associated with the use of real-world
applications for teaching hydrology to Geography and environmental science students are class activities (Yli-Panula et al., 2019) intended to provide experiential learning opportunities (Healey and Jenkins, 2000; Ives-Dewey, 2009) such as the use of data and computer programs for analysis of spatial phenomena (Bowlick et al., 2017), field trips (Krakowka, 2012; Lai,

1999; Schiappa and Smith, 2019), fieldwork (Elkins and Elkins, 2007; Mol et al., 2019; Ramdas, 2019), case studies (Hofmann and Svobodová, 2017) and guest lectures (Graham et al., 2017; Hovorka and Wolf, 2019). Experiential activities encourage critical thinking related to the environment (Hofreiter et al., 2007); increase an appreciation of landscapes and physical land surface processes (Karvánková et al., 2017); introduce role models that provide examples of career paths available for graduates (Solem et al., 2019); heighten an appreciation of sustainable practices (Robinson, 2019; Yli-Panula et al., 2019); and equip students with skills that improve marketability after graduation (Spronken-Smith, 2019). These activities diversify the skillset of students and thereby contribute to the training of future professionals equipped to address societal challenges related to water security and ecosystem management. Skill diversification allows for these professionals to contribute to: multidisciplinary problem solving where research teams from different branches of academia are required to search for solutions (Scholten et al., 2007); interdisciplinary activities involving a combination of knowledge approaches within a field of inquiry (Cosens et al., 2011); and transdisciplinary synthesis where new fields of inquiry are created by the combination of disciplines (Krueger et al., 2016). Multidisciplinary problem solving is required for conservation (Dick et al., 2017) and water security (UNESCO, 2019) due to the complexity and heterogeneity of environmental systems at a number of scales and the need for social issues to also be addressed in context with these systems. Interdisciplinary activities are often encountered in hydrology due to water resources management related to civil engineering, groundwater extraction, and water use, necessitating the consideration of non-stationarity and human activity as integrated within the hydrologic cycle (Vogel et al., 2015). Transdisciplinary synthesis is required for the production of water knowledge between stakeholders, governments and academics to find innovative solutions to water issues that have been influenced by the philosophies and methodologies of traditional fields of inquiry (Krueger et al., 2016) and can involve data collected using techniques traditionally associated with different disciplines (Rohde et al., 2019).

Effective hydrology education requires knowledge synthesis to encourage progress in the hydrological sciences (Wagener et al., 2007). However, hydrology students often have a diversity of academic backgrounds and aptitudes that create challenges related to the training of students with strong competencies in mathematics, physics, sociology, psychology, and fieldwork required for research and for finding solutions to cross-disciplinary hydrological problems (Seibert et al., 2013). To heighten student interest and increase the applicability of geographic place within the context of a hydrology class, homework and class assignments should be relevant to individual students and grounded in experiential reality related to local hydrological processes (Van Loon, 2019). Moreover, classroom activities should integrate new developments in hydrology and associated technological advances to better prepare students for working on environmental challenges over the course of a career, thereby mitigating a well-known issue in the field of hydrology where application of scientific advances are often not applied in an operational context. In addition, the practice and teaching of hydrology is often extensively associated with Western cultural perspectives that do not take into consideration the viewpoints and practices of local and regional cultures; instructors should therefore endeavour to represent ideas and conceptualizations from these cultures in active teaching practice (Ruddell and Wagener, 2015).

Instructors have formulated novel class activities to address the needs of hydrology education. For example, Kingston et al. (2012) describe a student activity involving the collection of GPS data using mobile computing devices and DVD technology used to implement a virtual tour of a weather station field site. Students processed the self-collected GPS data using a GIS system and the DVD also provided multiple choice questions for student self-assessment of comprehension. Van Loon (2019) developed assignments where students select a river for analysis. The students then subsequently completed homework tasks and formulated a poster presentation related to the associated hydrology of the area. Lyon et al. (2013) created a field course where students were able to propose, implement and document a self-directed program of data collection and analysis to characterize the ecohydrology of a Mediterranean location in Greece. To teach challenges related to sharing of water resources and associated conflicts, Seibert and Vis (2012) developed the "Irrigania" computer game to simulate "tragedy-of-the-commons" scenarios related to water use and farming between individual farmers and villages, whereas Hoekstra (2012) and colleagues developed the "River Basin Game" and the "Globalization of Water Role Play" game to teach elements of water management at regional and global scales. The use of simulation games in the classroom complements traditional methods of teaching and allows for experiential learning (Rusca et al., 2012). For all these class activities, students reported a greater satisfaction with respect to the learning experience and the development of skills useful for the solution of real-world problems. These activities are also good examples of a constructivist teaching approach.

Hydrology as a quantitative science requires data for characterizing hydrological processes and associated phenomena. The development of innovative technologies for distributed spatial and high-temporal resolution data collection in association with the archiving and analysis of large datasets is essential for driving advances related to the understanding and modelling of hydrological processes (Tauro et al., 2018). Projects such as the WMO HydroHub (https://hydrohub.wmo.int/en/home) (Dixon et al., 2020) and EnviroDIY (https://www.envirodiy.org/) allow for sharing of devices, data and measurement projects between scientific collaborators and stakeholders, thereby supporting United Nations SDGs. In a similar fashion, data availability is an essential component of hydrological education at the post-secondary level and necessary for advancing the science of water education (Ruddell and Wagener, 2015).

However, in the context of a traditional hydrology course, datasets such as water level, velocity and associated discharge are often collected from streams using standard field techniques such as wading rods and current flow meters. These example datasets are not extensive and are therefore limited in a spatial and temporal fashion, particularly when students are briefly taken to a field site for a data collection opportunity. Student experimentation with data collection instruments only occurs during the time of the field trip and the students do not borrow instrumentation for private experimentation. This can be considered as a lack of data availability for curiosity-driven learning experiences and construed as a type of data scarcity. Moreover, plots of actual data collected at field sites are not the same as synthetic curves in textbooks and therefore students should be exposed to the collection of actual data. Data often requires imputation, averaging or filtering prior to providing inputs for hydrological models (Gao et al., 2018) and some hydrology textbooks do not present this in a clear fashion. Students who have worked with actual data appreciate the nuances and challenges of

datasets (Lim et al., 2020) and are better prepared for graduate school research and environmental science jobs (Hovorka and Wolf, 2019).

Electronic circuits are deployed at field sites to autonomously collect data (Hund et al., 2016; Navarro-Serrano et al., 2019; Wickert et al., 2019) and are used to provide inputs to mathematical models for hydrological prediction and forecasting (Lavers et al., 2020). When visiting geographic locations equipped with electronic instrumentation, students often appreciate the presence of meteorological stations that collect precipitation, solar radiation, heat and energy balance data. These stations can provide data useful for class assignments. The stations may have also been used for the creation of figures in research papers and lectures that the students have read and utilized. Consequently, students are aware of the need for electronic instrumentation to measure hydrological processes. Despite working with data collected by electronic circuits at field sites, students at the postsecondary level are often not exposed to how these circuits are calibrated or why these circuits work. Calibration is important for circuit operation where an output is corrected for accuracy (Kouider et al., 2003) or related to a physical quantity. A classic example of calibration involves pyranometers to measure solar radiation from the sun: an output voltage is related to a solar radiation flux (Faiman et al., 1992; Kim et al., 2018; Norris, 1973). In addition, hydrological models are often calibrated to select the best model parameters to represent a physical reality (Gupta et al., 1998; Singh and Bárdossy, 2012). Circuit calibration and hydrological calibration can be considered as similar processes since both require the discovery of a transfer function that relates a set of inputs to outputs. Since students may be required to calibrate hydrological models or electronic circuits as environmental science professionals, students should have the opportunity to learn these important skills in the classroom. When an environmental sensing circuit fails to operate as expected, students should understand that re-calibration is required or that the circuit is not collecting valid data. Moreover, understanding circuit operation may motivate students to eventually develop novel and interesting environmental monitoring circuits, thereby encouraging the development of innovative sensors that help to provide better insight into environmental phenomena.

The open source electronics movement has reduced the cost of monitoring environmental phenomena, democratizing the use of instrumentation and providing non-proprietary methods for data collection that do not require the use of expensive software licenses for data access and programming (Pearce, 2013). This electronics movement is also associated with the concept of Open Data (Borgesius et al., 2015) and FAIR (Findable, Accessible, Interoperable, and Reusable) data (Mons et al., 2017) since the hardware and software of the instrumentation used to collect observations can be easily analysed due to availability of schematic diagrams and design files, thereby allowing for an unprecedented understanding of file formats, metadata produced during a data sampling process, and device operation that provides insight into the accuracy and precision of the sampling procedure. More specifically, using the FAIR acronym, the data is Findable due to the potential for automated upload to online databases; Accessible since file formatting is known and the data is available for download; Interoperable since the data can thereby be utilized with other datasets; and Reusable to be employed in the context of more than one study in a fashion that also encourages replication of experiments. Examples of systems built on open source technology that utilize these principles include 4ONSE (4 times Open and Non-Conventional

technology for Sensing the Environment) (Sudantha et al., 2018) and the ODM2 Data Sharing Portal (Horsburgh et al., 2019) designed for interoperability with open source hardware. Electronics designs using elements of open source technologies often include the use of the Arduino platform to create low-cost dataloggers (Hund et al., 2016; Wickert et al., 2019). The introduction of custom electronic circuits in a hydrology class in lieu of commercial devices exposes students to the idea that instrumentation can be developed locally without a high cost of acquisition. Given the opportunity, students can build and test circuits used for environmental measurement. This opportunity can be viewed as empowering and enriches the educational experience by allowing students to move beyond the nuanced idea of electronic environmental measurement circuits as black boxes, thereby enabling a better understanding of how these devices are constructed and calibrated.

Postsecondary hydrology classes do not often include the construction, testing and calibration of electronic circuits as an official part of the curriculum. Therefore, a research gap exists and there is a need to document these electronic circuit teaching experiences at the postsecondary level. Documentation of experiences provides other instructors with the information required to replicate these activities as well as assessment of the activities in the context of teaching and learning. Given these considerations, the objectives of the paper are to:

- Share the circuit construction, calibration, and data analysis experiences—including challenges, tips and resources—of the 2019 Advanced Hydrology (GEOG 427) class in the Department of Geography and Planning at the University of Saskatchewan (USask). The goal of the class is to introduce students to instrumentation, mathematical models, and hydrological processes, along with concepts of systems theory and critical thinking skills.

- Assess and highlight the benefits of integrating hands-on exercises utilizing simple technology into the hydrology post-secondary teaching curriculum to enhance learning using electronic circuits that allow students to experiment with instrumentation used for environmental data collection.

**2 Methodology**

A workflow diagram (Fig. 1) shows the stages involved in the class activity coincident with each of the sections of this paper. The stages are classified into time periods related to pre-activity, during-activity, and after-activity tasks. The activity was first proposed for departmental approval (Section 2.1) before circuit design, setup, and purchase of materials (Section 2.2). Then, technical references and circuit theory as discussed in the Supplement were prepared for student use (Section 2.3). Implementation of the activity (Section 2.4) was explicitly planned before circuit activity kits were given to the students; this planning involved a framework for collection of written feedback that includes the use of a conceptual model of learner-instructor feedback loops (Section 2.5). Section 3 discusses a systems approach associated with hydrological modelling concepts covered in class. The systems approach is also exemplified by electronic circuits as

collections of sub-systems. Section 4 describes the circuit activity as conducted in the class, indicating: the theory, construction, and circuit operation (Section 4.1); elements of classroom application (Section 4.2); and example applications for the electronic circuits identified by the students (Section 4.3), including the use of mathematical models (Section 4.4). After the activity, student open-ended feedback (Section 5.1) and closed-ended questions (Section 5.2) were analysed using the framework established in Section 2.5 to identify conceptual feedback loops (Section 5.3). Based on this classroom

activity, conclusions were identified (Section 6) indicating that there is a need for improvement of post-secondary hydrological education using innovative strategies and institutional support. The flowchart also indicates that there is a potential for possible implementation of the activity in other classrooms.

## 2.1 Course Activity Proposal and Acceptance

The course activities were proposed to the Department of Geography and Planning at USask six months before the start of the class in September. The activity proposal indicated the need for the course activities to include components necessary for teaching hydrology as summarized in Section 1 of this paper. Three months before the start of the class, the activities were approved along with a budget used to procure required materials.

## 2.2 Technical Setup and Material Purchase

Printed Circuit Boards (PCBs) were produced from digital design files designed specifically for this activity (cf. Section 7 and the Supplement). PCBs are designed using layout software utilizing layers in a similar fashion to a GIS system (Fig. 2). Although it is possible for an instructor to manufacture a class set of PCBs utilizing chemical processes and manual drilling (Andrade, 1965), this is a labour-intense process that is prone to human error and therefore it was easier and

185 cheaper to purchase PCBs from a "board house" supplier that created a batch of custom PCBs using automated manufacturing methods for a relatively low cost. PCBs can take weeks to months to arrive based on the supplier and the "turn time" associated with an order that affects the overall cost of the PCBs.

Circuit components including the PCBs are provided to the students individually bagged and labelled for the water detection sensor and the RH/air temperature and pyranometer sensors (Fig. 3a). Each individual bag is marked with a PCB

designator corresponding to a component or components on the circuit board. The designator is also marked on the circuit board using "silkscreen" lettering by the PCB board house supplier as specified by the design files. The use of designators allows students to expeditiously populate components on the PCB without the use of an associated placement diagram. There are three kits in total: a base kit with a microcontroller and power supply components that are common to the project, whereas a choice can be made related to which of the other kits to populate for a system used to measure RH/air temperature

and a pyranometer sensor to measure solar radiation. Five toolboxes of battery-operated soldering irons, pliers, scissors,

tape, screwdrivers and IC pin straightener tools were also provided to a maximum of 20 people allowing for approximately four students per group. The number of kits were limited due to cost and the initial size of the class (18 students).

The purchasing of materials for this activity required a month of preparation time since the components had to be obtained from an electronics supplier in larger quantities; individual plastic bags and a labelling machine were obtained from a packaging supply company; and most hand tools were sourced locally. Working with another person, the instructor had to allocate approximately one hour of setup time for the assembly of a single student kit. Although some electronic suppliers will provide components individually bagged as a kit if a bill-of-materials (BOM) is provided, the cost of a "kitting service" is often not justifiable if a small number of kits are required for a classroom activity.

The kits and additional components were placed in containers on a cart for transportation to the classroom (Fig. 3b). A dedicated lab was not required for this classroom activity since the instructor provided all required materials. Bags of additional components as well as additional tools, soldering irons and batteries were also provided in another container to ensure that damaged or lost components could be replaced during the activity. Moreover, additional tools ensured that students could more efficiently work together if some students required more time to use a tool during the construction process.

## 2.3 Preparation of Handouts and Guidance Material for Students

To provide background circuit theory for the instructor and interested students, reference books are recommended. Horowitz and Hill (2015) serves as an excellent starting point for learning electronic circuit design. Holdsworth and Woods (2002) is a good reference for logic gates. Williams (2014) provides an overview on how to program an Atmel AVR microcontroller. Similar books are also available for other microcontroller architectures. De Vinck (2017) offers a complete course on how to solder electronic circuits.

The schematics, circuit board design files and BOM are available as an electronic download associated with this paper (Section 7). All design files are licensed under the CERN Open Hardware Licence (OHL). Technical details related to circuit operation and calibration are explained in the Supplement. These details are provided for context and to provide students and instructors with background information so that the circuits are not perceived as black box systems with inputs and outputs. Context related to calibration is also important to guide students and instructors in the implementation of this activity. The schematics and calibration information are provided by the instructor to students as a printed handout or as a digital download from a class materials website or content management system (CMS).

## 2.4 Implementation of the Activity

Before circuits are constructed, the class is initially introduced to actual data collected from field sites. This actual data includes temperature, Relative Humidity and pyranometer observations intended to show similarities to the data

collected by circuits constructed in class. The students are responsible for calibration of these circuits. For a final class assignment, students are given the option to use the circuits to collect and analyse data to provide inputs for a hydrological model. The assignment is open-ended in that each student is responsible for constructing a simple hydrological model driven by data collected from the circuits that students constructed in class. In this manner, students learn about how the circuit operates and how a hydrological model is assembled from different equations with associated assumptions coupled together to provide numerical outputs.

Students are provided with PowerPoint slides and a lecture on circuit construction. Students are also told that the circuit construction activity contributes to a 4% overall participation grade in the class. This participation grade was also assessed throughout the term based on attendance and class participation and was not primarily associated with the circuit activity. The circuit construction activity is introduced as a type of "game" or "puzzle" comprised of matching components to designators on the circuit board and soldering the components in a correct place. This is a form of "gamified learning" that can improve the enjoyment of an educational activity (Subhash and Cudney, 2018).

Everyone in the classroom working on circuits must wear safety glasses to prevent eye damage from flying debris when component leads are trimmed after soldering; this also prevents injury from particulate matter during the soldering process. Students are also told about the possible hazards of soldering prior to the activity. Safe soldering techniques are demonstrated so that no injuries associated with burns or scratches occur during the circuit construction.

The activity was implemented so that students can help each other construct the circuits and formulate novel methods for circuit calibration and application. Students bend, insert and solder components into holes on printed circuit boards (PCBs). Students are encouraged to work together on the associated calibration and modelling assignment, although each student is responsible for submitting an individual write-up. This is a form of cooperative learning that is known to enhance retention and engage students (Slavin, 1980).

Students are given handheld temperature, RH, and solar radiation flux meters to serve as calibration references. These handheld devices are commercially available and are listed in a downloadable BOM associated with this paper. Calibration depends on the type of sensor populated by each student on the PCB (temperature/RH or pyranometer). For handheld commercial meters that reported light outputs in lumens or lux, students use an appropriate conversion factor to obtain a solar radiation flux in units of $W\,m^{-2}$. Further details on calibration are given in the Supplement.

## 2.5 Design, Analysis and Collection of Written Feedback

Student-provided feedback was used to assess the efficacy of this activity for teaching hydrology at the postsecondary level. The Narciss (2013) and Hattie and Timperley (2007) models provided a conceptual framework for the collection and analysis of feedback associated with this activity. The Wilson et al. (2014, p.76) framework was used to indicate that the circuit activity can motivate student interest.

The conceptual framework used for the design of written student feedback related to this class activity is outlined by Narciss (2013) as the interactive tutoring feedback model (ITF-model). The ITF-model incorporates ideas of systems theory and constructivist teaching. These two ideas are also utilized within the context of the student activity described in this paper. Narciss (2013) considers both the learner and the instructor as control systems within three feedback loops. One feedback loop is internalized by the learner and is related to student satisfaction, knowledge creation and retention, whereas another feedback loop is used by the instructor to adjust the class activities to achieve external competency standards and maximize learning associated with instructor and department-specified goals. The third feedback loop links the instructor and student systems together. The purpose of each conceptual feedback loop is to establish equilibrium in each system with respect to a setpoint that represents a goal or a level of competency. In this paper, the three feedback loops are identified as learner (L), instructor (I), and learner-instructor (L-I) to identify the system domains where these feedback loops operate.

Quantification of learning is to understand how these feedback loops are operating. To quantify each of the three feedback loops identified by Narciss (2013), the Hattie and Timperley (2007) model is used to construct questions. According to Hattie and Timperley (2007), feedback questions are associated with four levels. These four levels are associated with each of the feedback loops in the list below. Although all feedback loops are interrelated, each of the levels have been matched to the feedback loop that is mostly indicative of the level:

1. Task Level, associated with how well the task is performed in relation to an external standard (L-I and I).
2. Process Level, related to how learners perceive the tools and techniques of a task (L, L-I, and I).
3. Self-Regulation Level, indicating a self-assessment of how well a student performs the task (L).
4. Self Level, related to the change in self-worth of a student engaging in the task (L).

Open-ended feedback allows for all the task levels and functioning of the conceptual feedback loops to be assessed. This is because open-ended feedback solicits responses without imposing a more rigid structure associated with specific task-based questions (Ballou, 2008). Students in the class were voluntarily asked to provide open-ended written feedback on the circuit activity for assessing quality of classroom instruction (Davis, 2009, p.461–463). An open-ended feedback sheet was handed out in class. Eleven students returned the open-ended feedback sheet. The question asked at the top of this sheet was "Using the space below, please provide your thoughts and feedback on the circuit activity." The students signed a permission consent to indicate acceptance associated with using the feedback in a published paper.

Class feedback in a more structured format involving rating questions and written feedback was also solicited by the College of Arts and Science at USask as the Student Learning Experience Questionnaire (SLEQ). Data from the end-of-term SLEQ was collected using the *blue* experience management feedback system (provided by *explorance*, https://explorance.com/products/blue/) in accordance with policies established by USask. This automated feedback system aggregates and anonymizes responses. All responses from students are voluntary and eleven students in the class completed this online survey. The SLEQ questionnaire allowed for all four task levels to be assessed.

It is not possible to determine whether the same students also completed the in-class open-ended written feedback sheet. Responses could not be matched to individual students. The SLEQ data is also provided as an associated download for this paper. The class instructor personalized some of the questions asked by the SLEQ survey. Along with standardized closed-ended questions asked by the Department of Geography and Planning and the College of Arts and Science at USask, these personalized open-ended questions helped to address the efficacity of teaching and learning associated with this circuit activity. The questions below are classified to indicate the targeted task level. The questions cover all four task levels to assess the three conceptual feedback loops.

1. What are your thoughts about the building, construction, and calibration of electronic circuits in this class? (Process, Self-Regulation)
2. Do you think that calibration of circuits and understanding how circuits work are important learning experiences for environmental scientists? Why or why not? (Task, Process, Self-Regulation, Self)
3. After taking this class, how do you perceive systems theory as being important in environmental science? (Task, Process)
4. What did you learn about critical thinking in the class related to the use of electronic circuits, models and hydrological processes? (Task, Process, Self-Regulation)

To identify trends and similarities in the feedback, the Voyant web-based tool (https://voyant-tools.org/) was used for quantitative text analysis (Rockwell and Sinclair, 2016). Student feedback responses are provided as a download associated with this paper. The responses were transcribed and anonymized without modification of grammar or sentence structure since this maintains context and intent.

## 3 Systems Approach and Hydrological Modelling

### 3.1 Hydrology as a Systems Science

Students in the class are initially introduced to the concept of hydrology as a systems science. A systems thinking approach is important in geographic education since this enhances understanding of spatial relationships and interconnections between processes (Cox et al., 2019a, 2019b). At the postsecondary level, systems thinking is a key component of courses designed for sustainable development competencies (Schuler et al., 2018). A system can be represented as a collection of entities that have attributes and an internal state (Hieronymi, 2013). The boundaries of the system are often demarcated as a type of control volume and visually represented by lines on a diagram (Fig. 4a). A system can have inputs and outputs. Sub-systems can be combined to form a larger system by coupling the outputs from one sub-system to the inputs of another sub-system. Examples of well-known systems with inputs and outputs include a vehicle, plumbing in a building, a computer and a smartphone as circuits, and the human body. Students tend to understand examples couched in experiential reality (Oliveira and Brown, 2016) and therefore these system concepts have the potential

to be more easily understood when presented to the class. Hydrological processes can also be represented as systems. A watershed is construed as consisting of basins and sub-basins that respectively correspond to systems and sub-systems with inputs and outputs (Zävoianu, 1985, p.9–25) (Fig. 4b). The sub-systems are coupled together by linkages indicative of water flowpaths associated with processes such as runoff, infiltration, evapotranspiration, sublimation, snowmelt, groundwater flow, and river discharge. The summation of inputs and outputs along with an appropriate positive and negative convention is the basic idea associated with the computation of a water balance using these sub-systems (Berghuijs et al., 2014).

### 3.2 Hydrological Modelling

Students learn about three different types of hydrological simulation models in the context of the class: physical, mathematical and analog (Dingman, 2015, p.597). A physical model is a representation of the world at a different scale; examples include tabletop watersheds, sprinkler plots and hydraulic structures with inputs and outputs of water. Mathematical models have state variables and transfer functions that represent equations coupled together to form inputs and outputs. Analog models nominally use concepts from electrical engineering and circuit theory to represent hydrological phenomena such as groundwater flow and evapotranspiration and have inputs and outputs. Described in this fashion, the three types of models are considered as systems.

Analog models are often implemented as mathematical models with equations evaluated using a computer program in lieu of building actual circuits (Ménard et al., 2014) (Fig. 5a). Actual circuits were constructed in the past consisting of elements such as resistors and capacitors (Sander, 1976; Shen, 1965; Skibitzke, 1960) (Fig. 5b,c) since computation resources were more limited in the 20$^{th}$ century and building of a circuit allowed for current or voltage to be measured and easily related to the magnitude of a hydrological process.

By conceptualizing hydrological models as systems with inputs and outputs, students can learn how to construct a mathematical model by combining equations along with assumptions in a similar fashion to a circuit, where circuit elements are combined to provide a set of outputs given inputs (Kang, 2016). The practice of assembling a circuit provides students with an experiential example of how sub-systems are combined to form a system.

A similar process also occurs when combining equations to compose a hydrological model. If equations are conceptualized as sub-systems with assumptions, students may find it easier to select equations, combine equations together, and understand how the inputs affect the outputs. This process is useful to learn since it is often performed by graduate students and hydrology research professionals who model hydrological phenomena using mathematics and associated computer programs to produce predictions and forecasts related to phenomena such as drought (Mishra and Singh, 2011), flooding (Teng et al., 2017), geotechnical slope stability (Fawaz, 2014) and avalanche activity (Morin et al., 2020).

Students are given the opportunity to propose their own mathematical models for use with electronic circuits in lieu of everyone in the class being obligated to complete the same assignment. This is in line with the concept of constructivist teaching and learning. Moreover, allowing students to be creative enhances geographic learning (Yli-Panula et al., 2019).

Applied to an environmental science classroom, this instructional method can increase class exam scores and student
satisfaction ratings (Lord, 1999).

## 4 Electronic Circuits

### 4.1 Theory, Construction and Circuit Operation

Since the class is not an engineering course, students do not design circuits since there is not enough time in one
term to learn circuit design and PCB layout.  The creativity aspect of the course in line with constructivist and experiential
learning is associated with the use of the circuits to collect data utilized as inputs to student-proposed mathematical models.
However, circuit theory and associated circuit operation (as described in the Supplement) is provided by the instructor for
learning context and relevance.

Students were given the opportunity to assemble (Fig. 6) and take home two circuits: a water detection circuit (Fig.
7) and a circuit with a relative humidity (RH)/air temperature sensor as well as a photodiode with a diffuser that is calibrated
as a pyranometer to measure shortwave radiation (Fig. 8).  The water detection sensor as the first project (Fig. 7) provides
students with an opportunity to learn some basic circuit theory and construction skills.  This first project is deliberately kept
simple.  This is because students learn best in a formal teaching environment where skills and concepts are presented in a
fashion progressing from simpler to more complicated formulations (Davis, 2009, p.280–281).  Since the second circuit (Fig.
8) can be constructed in three configurations using either or both RH/air and pyranometer sensors, this allows the students to
choose which configuration would be most useful for collection of data for the formulation of hydrological models.  A
microcontroller is used as a control system.  The temperature, relative humidity and calibrated solar flux is displayed on an
LCD display driven by the microcontroller (Fig. 8b, c).  This allows for the data to be easily read in lieu of utilizing other
instrumentation such as a voltmeter.  To acknowledge the ancestors of the land on which USask is built and to contribute to
Indigenous reconciliation in Canada (Castleden et al., 2017), the LCD display shows "Welcome" and "tawāw": words in
English and Cree indicating that the device is ready for operation (Fig. 8a).  The Cree word tawāw can be literally translated
as "come in, you're welcome; there's room" (Wolvengrey, 2001, p.218) and allows students to view words other than
English that are normally displayed on most LCDs.  The letter ā with a macron (overbar) had to be uploaded to the LCD
display memory since the default LCD character set did not support this special character used for the Cree Roman
Orthography.

**4.2 Classroom Application**

Construction of the water detection sensor took approximately 1.5 hours as the maximum class time and one class
was allocated for this activity.  Construction of the RH/air temperature circuit took approximately 4 classes for a total time of

6 hours. Out of this total time, three classes (4.5 hours) had to be allocated for construction, and one class (1.5 hours) for debugging of soldering errors associated with construction. Students who had soldered in the past verbally indicated during the activity that they found the circuits easier to construct than students who had not worked with soldering tools and techniques. Students who had previous experience were thereby paired with students who had less experience. This practice reduced tension and prevented students from feeling excluded. Indeed, three students verbally indicated to the instructor that they liked the idea of circuit construction, but they did not feel competent to successfully construct the circuits. The instructor alleviated the concerns of these students by assisting everyone in the classroom during the activity.

The finite number of tools allowed students to work together. While one student soldered a component, the other students in the group were able to observe the soldering process and learn from each other. Students were able to help each other with part placement and orientation. This process emphasizes the importance of using battery-operated soldering irons since students were able to quickly exchange and share irons without the disadvantage of having to deal with cords and power cables. However, a number of additional AA batteries had to be kept in the classroom for facilitating this activity since the batteries in each soldering iron lasted for approximately 45 minutes of continuous operation and had to be changed in-class to allow students to continue working on the circuit for the total class time of 1.5 hours.

Students used the handheld temperature, RH and solar radiation flux meters for circuit validation and calibration. For meters that reported light outputs in lumens or lux, students used an appropriate conversion factor to obtain a solar radiation flux in units of $\mathrm{W\,m^{-2}}$. Some students took the circuit and hand-held light meters outside to measure solar radiation flux from the sun as an outdoor calibration procedure (Fig. 9a), whereas others used an indoor calibration procedure utilizing smartphone LED flashlights (Fig. 9b, c). This was a novel development not anticipated by the instructor. Alternately, some students used radiation flux data from a nearby meteorological station for sensor calibration.

Figure 9b and Fig. 9c show students engaged in the smartphone LED flashlight calibration procedure. The light intensity of the LED flashlight was measured using a handheld light meter. The voltage output of the pyranometer circuit was then determined using the LED flashlight. An associated calibration coefficient was subsequently obtained using techniques identified in the Supplement. The coefficient was refined using further experiments conducted at an outdoor location.

From this calibration procedure, the students found that:

- Pyranometer calibration did not work well at an indoor location such as in the atrium of the Agriculture Building at USask. Despite the presence of many windows in this location, sunlight that enters the building interior is directional and strongly diffused by metal and glass reflecting surfaces and is therefore not an accurate representation of solar radiation associated with a hemispherical sky. Pyranometers calibrated in this indoor environment did not provide an approximate representation of radiation flux at an outdoor location.

• Due to the inexpensive cost of the photodiode used for the pyranometer, calibration coefficients did not approximately coincide between PCBs due to manufacturing tolerances. Calibration coefficients could exhibit order of magnitude differences. Students realized that all photodiodes were not alike, and that calibration is important for this application.

• Temperature and RH as reported by the circuit (Fig. 8b) agreed to within $\pm 1\,°C$ and $\pm 3\%$ RH. This is approximately the same as the accuracy reported by the AM2320 datasheet used for the circuit and includes differences due to temperature gradients and boundary layer effects near the PCB.

• For the pyranometer (Fig 8c), the average error ranged between approximately 3% to 35% compared to handheld light/radiation meters. The error was dependent on calibration and the position of the PCB relative to the sun. For more accurate measurements, the pyranometer sensor had to be positioned level to
430 the ground to allow for an approximation of a hemispherical measurement of radiation.

Some of the PCBs constructed by the students had manufacturing defects. Table 1 shows the estimated number of defects that had to be repaired before student PCBs worked successfully. Soldering joint defects were the most prevalent. Another common defect was related to some components not placed with the right orientation. Since PCB repair and re-work tools were not available in the classroom, the instructor repaired the PCBs outside of class time.

Although most students constructed a hybrid RH/temperature/pyranometer circuit, no student chose to only construct a pyranometer (Table 2). Two students constructed a PCB with only a RH/air temperature sensor and did not populate the pyranometer components. For the RH/air temperature and pyranometer circuit, four classes were required to construct the circuit and students that did not attend the class on one day could attend on another day to work on circuit construction. The students that did not construct a PCB were either absent from class during circuit construction or auditing
the class. All students who constructed a water detection circuit were also present for the construction of the hybrid RH/temperature/pyranometer circuit.

**4.3 Example Applications for Electronic Circuits**

Students identified novel applications that were not discussed by the instructor. One student expressed an interest to use the water detection circuit to determine if seasonal water levels exceeded a certain height in wetlands, indicating
whether water could be exchanged between two nearby ponds. Another student suggested that the RH/air temperature circuit and pyranometer could be used in lieu of more expensive micrometeorological handheld instruments to collect measurements at a field site during fieldwork. The wetness detection sensor can be used as a groundwater dipper (Fulford and Clayton, 2015) to determine the level of the water surface in a groundwater well. The sensor would be waterproofed and fastened on the end of a wire for lowering down an observation well. The piezoelectric buzzer would provide audible
feedback when the water sensor reaches the position of the water surface. The distance to the water surface can be determined using a series of graduated markings on the wire.

**4.4 Modelling with Electronic Circuit Data**

This section briefly summarizes possible models for classroom application. Students can use some of these models for a class assignment. Examples listed in this section were utilized, identified, or considered by students in the class, but other example applications might also be suitable based on regional geography if this activity is to be replicated at another location. Students are graded on how assumptions are provided in the assignment, since each equation coupled together to form a hydrological model is associated with assumptions. The students are also graded on how they interpret the model outputs. Students justify if the outputs are physically reasonable based on a literature search or on theoretical calculations providing insight into the magnitude or frequency of hydrological processes.

The RH/air temperature circuit can be used to obtain daily mean air temperature. The mean air temperature is related to snowpack temperature or snowmelt using an empirical degree-day or similar relationship for cold regions where snow is prevalent (Fontaine et al., 2002). Relative humidity and air temperature can also be related to transpiration processes (Clum, 1926; Mahajan et al., 2008). If the pyranometer circuit is used to measure incoming shortwave radiation from the sun, outgoing shortwave radiation can be estimated using an assumed albedo (Oke, 1992, p.86). The longwave radiation can be obtained from a semi-empirical (Sicart et al., 2006) or physical model (Viúdez-Mora et al., 2015). If sensible and latent heat fluxes are modelled or measured along with appropriate assumptions (Essery, 1997; Marks et al., 2008), the energy required for snowmelt and the resulting change in Snow Water Equivalent (SWE) can be calculated (Pomeroy and Brun, 2001, p.92).

**5 Feedback**

**5.1 Open-Ended Feedback**

Out of the 15 students who completed the in-class circuit activities, 11 students submitted an open-ended feedback sheet, indicating that four students completing the activity chose to not provide in-class feedback or were absent from class. Written feedback was mostly positive and indicated that students found the learning experience to be an enjoyable, informative, beneficial, and novel opportunity (Table 3). Three students indicated changes should be made to the activity: (1) more soldering irons required per group and the calibration process should be explained in a more in-depth fashion; (2) feet should be provided so that the PCB can be situated on a table; and (3) the activity took too long.

As indicated by Student #2, additional soldering irons should be provided to allow students to work more quickly, but this is dependent on the class budget and number of batteries available. Although battery-operated soldering irons are beneficial for allowing students to work freely without cords and cables, a permanent teaching lab for hydrological circuits would benefit from the use of AC-powered soldering irons to reduce the number of batteries required for this activity. The feedback response also suggests that the calibration process should be clearly explained in the context of systems theory and

mathematical modelling, so the instructor should be willing to address this concept in an in-depth fashion by extensive demonstrations, lectures or videos shared with the students to further enhance comprehension.

Since a student in the class recommended via written feedback that the PCB should have "feet" (Student #3) the instructor considered providing some feet for students who would like to glue the feet on the bottom of the PCB. Conventional plastic feet elevate the PCB above a surface and provide a convenient platform to situate the circuit while it is being operated. To engage students, the instructor used a 3D printer to create a class set of "feet" with toes that can be glued on the bottom of the PCB (Fig. 10a). Figure 10b is an example of a student standing next to a PCB with "feet." Injecting humour into a classroom situation is known to increase retention while engaging students by heightening interest and

attentiveness (Korobkin, 1988).

      One caveat addressed by student feedback was the time required for this activity (Student #7). Since the class did not have a lab section, classroom lecture time had to be used for this circuit-building activity, reducing the time for lectures and other activities. Future circuit construction activities should therefore be conducted in the context of a scheduled lab to allow class time to be effectively utilized for questions, problem-solving activities, and lectures.

The trends plot from the Voyant software (Fig. 11a) shows that there are five re-occurring words in the open-ended student feedback (Table 3): circuit, activity, building, fun and class. This demonstrates that students recognized the experiential aspects of the circuit building activity that occurred in class, and the activity was fun. Fig. 11b shows that most links in the student feedback are associated with the following words: building, activity, and circuit. Although the word "fun" is linked with "circuit" and "activity" the figure demonstrates that most feedback was focused on the circuit

construction. The linkages between the rest of the words indicate that the kits allowed for the incorporation of a unique activity into the class that gave an opportunity for students to experience the building of a circuit and experiment with the circuit for data collection. This interpretation is further supported by the associated mandala plot (Fig. 11c) that shows most students found the circuit activity to be "great" and "fun" although "experience" and "opportunity" are also a part of the responses. The word "learning" or "learnt" shows up in the responses of four students (Student 1, 3, 5, 11) indicating that

this was not explicitly identified by most students as a goal of the activity; however, the learning process and engagement of students is implicitly demonstrated by the responses.

      Students identified that the circuit activity provided a useful insight into calibration and operation of electronic devices utilized for fieldwork (Student #1) and that it was "*cool to be able to have your own circuit*" for the collection of actual data (Student #4). Student #5 identified the process of circuit building as a "*great learning experience*" but indicated

that technical aspects of the building process were unrelated to hydrological processes. However, Student #10 indicated that the activity was "*...a great way to incorporate Hydrological Modelling and circuit building*" and Student #11 indicated that the circuit activity "*...allowed us to make correlations between what we learnt in class and apply it to an activity.*" Student #6 indicated that the circuit building activity "*helped me to understand how these sensors work*" and mentioned that the skills developed in class would have future use. Student #8 indicated that the circuit activity provided an incentive for

students to attend class, whereas Student #9 found that the circuit activity was unique, useful, engaging, and enjoyable.

Although the circuit activity was primarily intended to teach systems theory in hydrology and to allow students to collect data for use with mathematical models of hydrological phenomena while providing a background to the use of electronic circuits for environmental monitoring, there is a possibility to further develop this activity.  As suggested by verbal feedback from students after the activity, the circuits should be modified to log data to flash memory (such as SD cards), thereby enabling automated collection of data.  Students also verbally indicated that the plastic jumpers used to set calibration coefficients were challenging to insert and remove.  Moreover, at least two students remarked that the PCBs should be waterproofed to allow for usage in wet environments such as at a field site location.  Although increasing the complexity and cost of the circuit construction activity, these changes would be worthwhile additions to a circuit used for teaching purposes.  Since environmental scientists and hydrologists often utilize electronic circuits in conjunction with dataloggers, circuit design could also be introduced to advanced undergraduates and graduate students in the context of a separate class at the undergraduate level.  This class would enable students to propose, design and test their own electronic circuits, thereby encouraging innovation and allowing students to develop important skills useful for the collection of data in the context of graduate work and environmental consulting.

## 5.2 SLEQ Feedback

The SLEQ feedback closed-ended questions allowed for a quantitative analysis of student responses submitted directly to the university.  The feedback was not submitted to the instructor and was processed using a third-party web survey service in accordance with university policies.  Using "*A great deal*" or "*mostly*" as rankings, the respondents indicated that the class provided a "*deeper understanding of the subject matter*" and in this regard, the class responses associated with number of students per ranking scored slightly higher than other classes offered by the department and college at the same 4th year level.  Also, in the same fashion, the respondents found the class to be "*intellectually stimulating*" and once again the scores are slightly higher compared to other classes.  Most respondents indicated that the class projects and assignments provided opportunities to "*demonstrate an understanding of the course material.*"  The quality of learning in the class was rated as "*Excellent*" or "*Very good*" by all respondents.

Although the class did not have a separate lab section, the SLEQ feedback form also asked the students questions related to the quality of experiential learning in the context of a lab that includes the circuit activities.  Although most respondents indicated that the course lab circuit activity provided transferable skills for other courses with a "*A great deal*" or "*mostly*" rankings, two students provided a lower ranking at the "*Moderately*" level.  The quality of learning experience in the lab component associated with the circuit construction was ranked "*Excellent*" or "*Very good*" by most respondents, although one respondent indicated that the experience was "*Good*."  For these questions, the rankings either coincided with or exceeded the rankings associated with other classes at the department and college level.

The written feedback provided by the SLEQ survey indicates that the circuit construction class experience was novel and unique, but in a similar fashion to the the open-ended feedback, some respondents indicated that it took longer

than expected, indicating that a dedicated lab section might be useful to hold in conjunction with the class. The student feedback indicates the novelty of the activity. In the words of one student, "*I never thought I would be building a circuit within my university journey, but I am happy to have experienced it.*" All respondents agreed that calibration and learning about how circuits work is an important educational experience for environmental scientists. Most respondents indicated that they found systems theory to be important, although one student indicated "*...still confused by it, I think it is good knowledge to know.*" Most respondents also indicated that they enjoyed the activity and one student indicated that "*I would recommend continuing the activity in upcoming classes, as it is something unlike any other class.*" Some students also commented on the exploratory nature of the course and the emphasis on not having a right or wrong answer when students had to develop their own mathematical models. In the words of one student, "*I had to develop more critical thinking skills as the assignments didn't really have any exact right answer*" whereas another student commented that "*circuit building, 'story telling' with mathematical equations, building and organizing models, systems and sub-systems*" were skills developed in the context of the course.

### 5.3 Feedback Loops

This section identifies the feedback loops and the Hattie and Timperley (2007) levels associated with the feedback. The feedback loop model is related to the Wilson et al. (2014, p.76) framework for motivating student interest in the classroom.

**I Feedback Loop**: Defects listed in Table 1 could have been mitigated by having the students watch a video demonstrating circuit assembly from a first-person perspective (Fiorella et al., 2017). Although the instructor walked around the classroom and demonstrated soldering and assembly skills during the circuit construction activity, students learned skills from a third-person perspective. These instructor demonstrations may have been less effective than if a first-person perspective video was used in a pedagogical fashion. The act of the instructor repairing the PCBs outside of class time agrees with the philosophy of constructivist teaching where the teacher acts as a facilitator of learning.

Before beginning the activity, the instructor had a preconceived notion that most students in the class would construct a RH/temperature circuit and that only a small number of students would choose to also populate the parts for a RH/temperature/pyranometer circuit. This unfounded thought was based on the idea that most students will choose a simpler project to reduce required effort and maximize gain. But as identified by Inzlicht (2018) as the "Paradox of Effort," more effort expended when completing a task can serve to increase the perceived cognitive value of the task, despite physical activity being costly in terms of physiological energy use.

**L Feedback Loop, Self and Self-Regulation Levels**: Psychological studies of student motivation in higher education show that students are willing to attempt tasks based on an intrinsic motivation that varies between individuals and a classroom environment cultivated by the instructor where students feel a sense of belonging and interest in performing the activity (Nupke, 2012). Other common factors for motivation that may have played a role include novelty of the activity as

per self-determination theory (González-Cutre et al., 2016); a participation grade to serve as an incentive for class participation (Czekanski and Wolf, 2013); the possible usefulness of the circuit in a class assignment as necessary for completion of the course; and a perception that participating less in the activity would not provide an experience justifying the monetary cost of taking the class, particularly since the circuit was a tangible item as a type of "self-gift" taken home after construction. A "self-gift" may boost self-esteem and encourage an individual to recognize "specialness" (Mick and

Demoss, 1990). Moreover, the construction of the circuit is an achievement and taking home a tangible and functioning circuit can be construed as a type of badge or acknowledgement. Badges function as awards, increasing social self-worth, and thereby serve as incentives for behaviour (Ling et al., 2005).

**L and L-I Feedback Loops, Process and Self-Regulation Levels**: Similar wording was used by the respondents who completed the instructor-provided and open-ended sections of the SLEQ evaluations, indicating that both feedback

forms served as reasonable assessments of the class and circuit activity. However, since the open-ended feedback form was given directly to the instructor, students used this form to provide more implementation-based suggestions for improving the circuit activity than the SLEQ evaluation that served to provide a final assessment of class feedback at the end of the term. The SLEQ evaluation was also the most anonymous. These anonymous types of feedback forms make it more likely that the students will provide a less-biased assessment of the class (Stone et al., 1977), although complete anonymity may not always

be reliable nor accurate due to lessened accountability (Lelkes et al., 2012). Therefore, both the instructor-provided feedback and SLEQ evaluation can be used together to obtain a more comprehensive understanding of how the students assessed the circuit activity.

**L Feedback Loop, Process and Self Levels**: The open-ended feedback shows that the students provided responses indicating that the circuit construction activity was grounded in experiential reality; to the students, this was a "hands-on"

activity that was also fun. Most student respondents did not use the word "learning" in the open-ended feedback solicited by the instructor, although this word appears in the SLEQ evaluation responses, suggesting that students identified the activities as more enjoyable than classic lecture-based forms of learning and that the learning process was implicit in the circuit building activities.

**L Feedback Loop and Self Level**: Class activities involving unique experiential opportunities engages students,

increases comprehension and facilitates learning (Gavillet, 2018). An element of enjoyment as "fun" for adult learners also heightens appreciation of the class and the associated subject material, increases retention of concepts, and leaves students with a positive and enjoyable experience of the learning process (Lucardie, 2014). The results thereby show that the use of circuits served to bring these elements into the context of the class.

**Process and Self Levels**: The SLEQ feedback standardized questions quantitatively showed that students in the

class perceived the educational experiences as leading to a better understanding of the subject area; to the students, these experiences were engaging and perceived to be intellectually beneficial. The instructor of the class was identified as facilitating learning and acting as a consultant for the student learning experience. These results underscore the efficacy of the constructivist teaching philosophy used for the class and emphasizes how this philosophy engages students and facilitates

learning. Since the standardized question scores for the class tend to exceed the average scores at the college or
departmental level, this indicates the usefulness of these techniques for teaching hydrology at the university level.

Despite the emphasis on circuit construction, analysis of written SLEQ feedback from the students in the class also identified that a "different" learning experience was offered in a fashion that is helpful for thinking about models and systems as conceptualizations of hydrological processes. Associated with the use of models were two related ideas of assumptions and calibration. Although these ideas are often used with mathematical models, the student responses did not
show a strong association with mathematics (Fig. 11 and Fig 12). This is important since the students learned concepts in class related to identification and analysis of models and systems. The use of mathematics was de-emphasized in favour of concepts, although the students used mathematics to propose hydrological models. De-emphasizing mathematics and focusing class lectures on concepts may have reduced any possible math anxiety present in students at the postsecondary level (Núñez-Peña et al., 2013), thereby allowing students to focus on gaining an understanding of hydrological and
environmental processes within the context of the class.

**Self Level**: The open-ended feedback (Table 3) did not provide any results related to the "gamified learning" aspects of circuit construction. There is a possibility that the idea of circuit construction as a game or puzzle enhanced student enjoyment of class activities, but since no questions were provided on the feedback forms explicitly addressing this concept, it is possible to only conjecture that the idea was beneficial to help facilitate the activity. Further classroom
implementation and analysis would be required to quantitatively assess the benefits of gamified learning to teach hydrology, particularly since computer games have been used to teach water resource management (D'Artista and Hellweger, 2007; Seibert and Vis, 2012), and further analysis may indicate the possible benefits of involving hydrology students in gamified learning opportunities. The constructivist teaching philosophy associated with this course was also demonstrated by the open-ended nature of the assignments where students were given the opportunity to propose models of environmental
phenomena. Allowing students to create knowledge helps to prepare students for future graduate and consulting work in hydrology and environmental science where mathematical modelling skills are necessary. The positive nature of the student feedback suggests that the circuit activity heightened interest in class subject matter and engaged students. Out of the total number of 18 students in the class, only three students did not participate in the circuit construction activities, indicating that 83% of students showed up for class and engaged in the activity.

**All Levels**: Wilson et al. (2014, p.76) indicate that techniques for motivating student interest in the higher education classroom involves "suspense, novelty, ambiguity, incongruity, or discovery." These terms can be used to analyse the student circuit activity and associated ideas related to systems and models in hydrology. The use of circuit construction is suspenseful since in a similar fashion to a puzzle, students assemble the circuit and after a successful assembly, the circuit can be turned on and data displayed on an LCD display. Also associated with this idea is the notion introduced by the
instructor as the circuit assembly serving as a type of "game" that is implicitly a puzzle. The circuit construction activity is a novel addition to a hydrology class. This statement is supported by student feedback, where some students indicated that they had not constructed a PCB and electronic circuit in the context of a class at the postsecondary level. Ambiguity and

incongruity are associated with the novelty of the circuit construction activity. Since many students had not constructed a PCB and electronic circuit, mistakes were made with respect to soldering and the placement of components. However, these mistakes can serve as useful learning opportunities where students are better prepared for future situations when an experiment does not work as expected (Glagovich and Swierczynski, 2004). Discovery was evident in the classroom since the students had the opportunity to formulate novel models of hydrological phenomena and to use a self-constructed circuit to collect data. These activities engaged students and allowed students to develop creative ways of thinking and learning.

## 6 Conclusion

The circuit construction and modelling activities described in this paper supported the implementation of constructivist teaching in a post-secondary undergraduate hydrology class. In the context of this activity, students constructed knowledge and did not serve as passive consumers of facts and materials provided by the instructor for rote memorization. The instructor also served in the role of a facilitator to support student learning. This motivated students to attempt the circuit construction task and heightened interest in the activity. Given the engagement of students by the circuit activity, similar constructivist teaching and innovative class activities should be more widely applied in hydrology classrooms at the post-secondary level to improve the student experience, enhance learning, encourage experimentation with electronic circuits for data collection, and allow for student-led creative problem-solving approaches to address educational challenges while subsequently training a new generation of scientists capable of applying knowledge from multiple disciplines to address societal challenges related to water. The lack of constructivist teaching and associated class activities at many educational institutions indicates the need for good teaching practices in hydrology at the postsecondary level. There is a need for universities to explicitly distinguish between a "lecturer" and a "researcher" to ensure specialization in teaching and thereby improve the student experience. Moreover, the design of courses by a joint team of lecturers and researchers in hydrology can ensure the use of well-founded teaching strategies conducted in relation with innovative classroom activities to meet the needs of students, and to also encourage the application of new methods, technologies, and developments in the hydrological sciences to meet SDGs and provide the foundation for a better collective future for the planet and for humanity.

## 7 Code and Data Availability

The microcontroller code, data, bill-of-materials (BOM) and associated circuit design files for replicating this activity are available as a link from Github (https://github.com/nkinar/Introducing-Electronic-Circuits) or Figshare (https://doi.org/10.6084/m9.figshare.12410588). This download also includes figures created by the Voyant software and anonymized student responses.

## 8 Author Contribution

All tasks (Conceptualization, methodology, software, validation, formal analysis, investigation, resources, data curation, writing—original draft preparation, writing—review and editing, visualization, supervision, project administration, funding acquisition) are contributed by N. K. Additional contributions to this paper are listed in the Acknowledgements section below.

## 9 Competing Interests

The author declares that there is no conflict of interest.

## 10 Acknowledgements

Aside from personal funding that I provided for this activity, I would like to acknowledge funding and support received from the Department of Geography and Planning and the College of Arts and Science at USask. Thank you to Department Head Dr. Alec Aitken and Director of the Centre for Hydrology Dr. John Pomeroy for supporting this activity. Funding for the Smart Water Systems Lab (SWSL) was received from Canada First Research Excellence Fund's Global Water Futures Program, the Natural Sciences and Engineering Research Council of Canada, Canada Research Chairs Program, and the Canadian Department of Western Economic Diversification (WED).

Permission was obtained to replicate Fig. 4a-b and Fig. 5a-c from other papers as identified in the figure captions. Thank you to John Wiley and Sons for providing the permission.

Ethics permission was obtained from the Department of Geography and Planning and the Research Ethics Board (REB) at USask. Thank you to Department Head Dr. Alec Aitken and the REB for approving this activity to be conducted. Ethics approval was granted by the REB as identification number BEH-2107.

Consent was obtained from students who are visibly identifiable in images of the classroom activity and from all students who participated in the activity. Also, thank you to the undergraduate and graduate students who participated in the activity (in alphabetical order): Kelby Brinley, Leah Clothier, Andrew Conan, Darin Duriez, Diane Eberle, Rebecca Kupchinski, Stefan Nenson, Danica Pittet, Mark Rambold, Jayvee Sadia, Maria Sanchez Garces, Nichole-Lynn Stoll, Ariel Thom, Josh Turtle and Michiel van Bemmelen. Another thank you is to be extended to former student Maria Stamatinos and PhD student Eric Neil (Department of Soil Science, College of Agriculture and Bioresources) who participated in the activity to build PCBs.

Thank you to an Anonymous Referee #1, Nilay Dogulu (Referee #2) and editor Stephanie Zihms for contributing many useful ideas and references that greatly improved the structure, presentation, content, and relevance of this paper. Everyone listed in this Acknowledgements section helped to facilitate the activity and for this I am grateful. kinanāskomitin

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

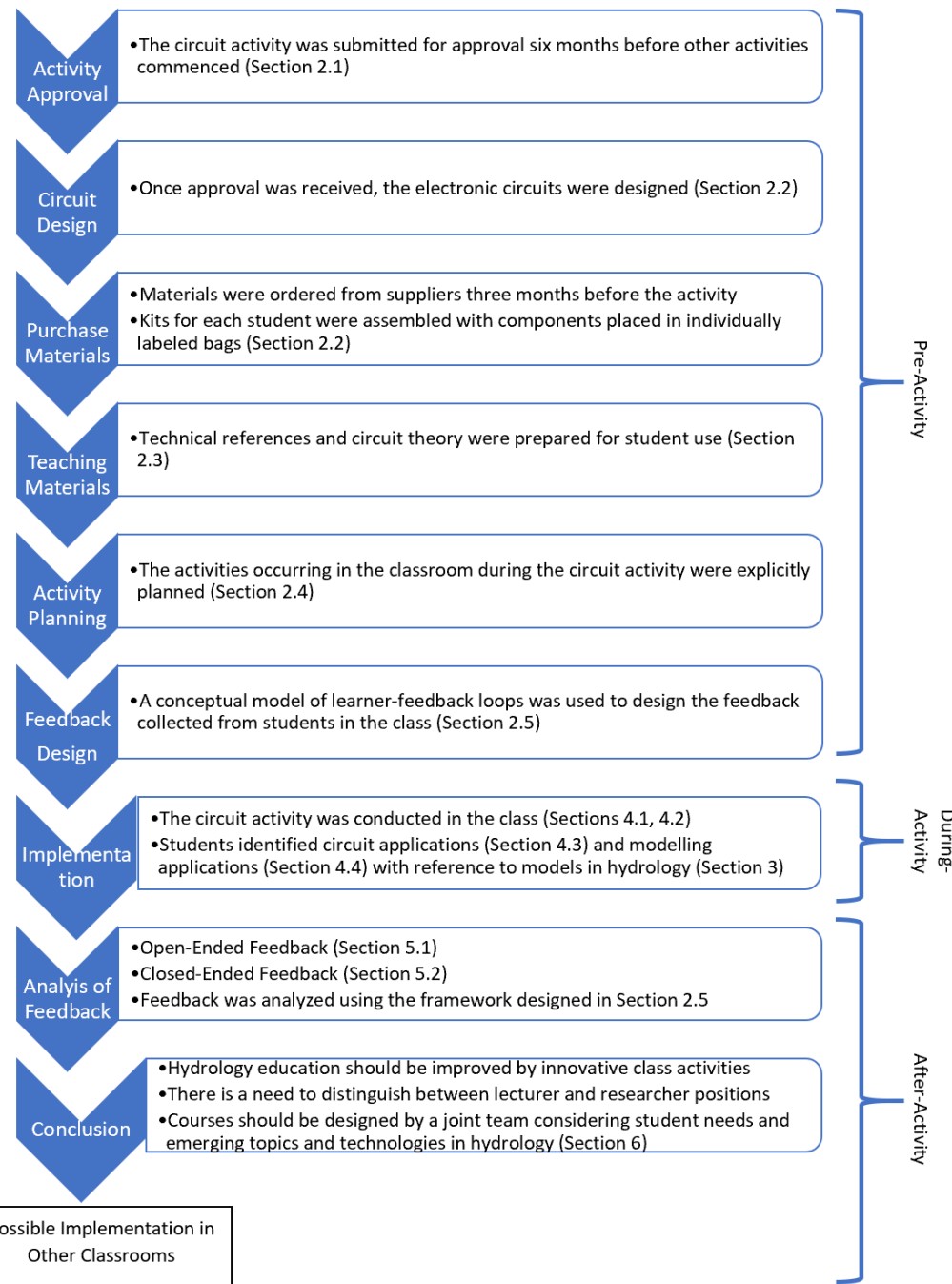

**Fig. 1**. Workflow diagram showing stages of the class activity. A temporal classification of stages according to pre-activity, during-activity and post-activity is shown along with a brief description of each stage.


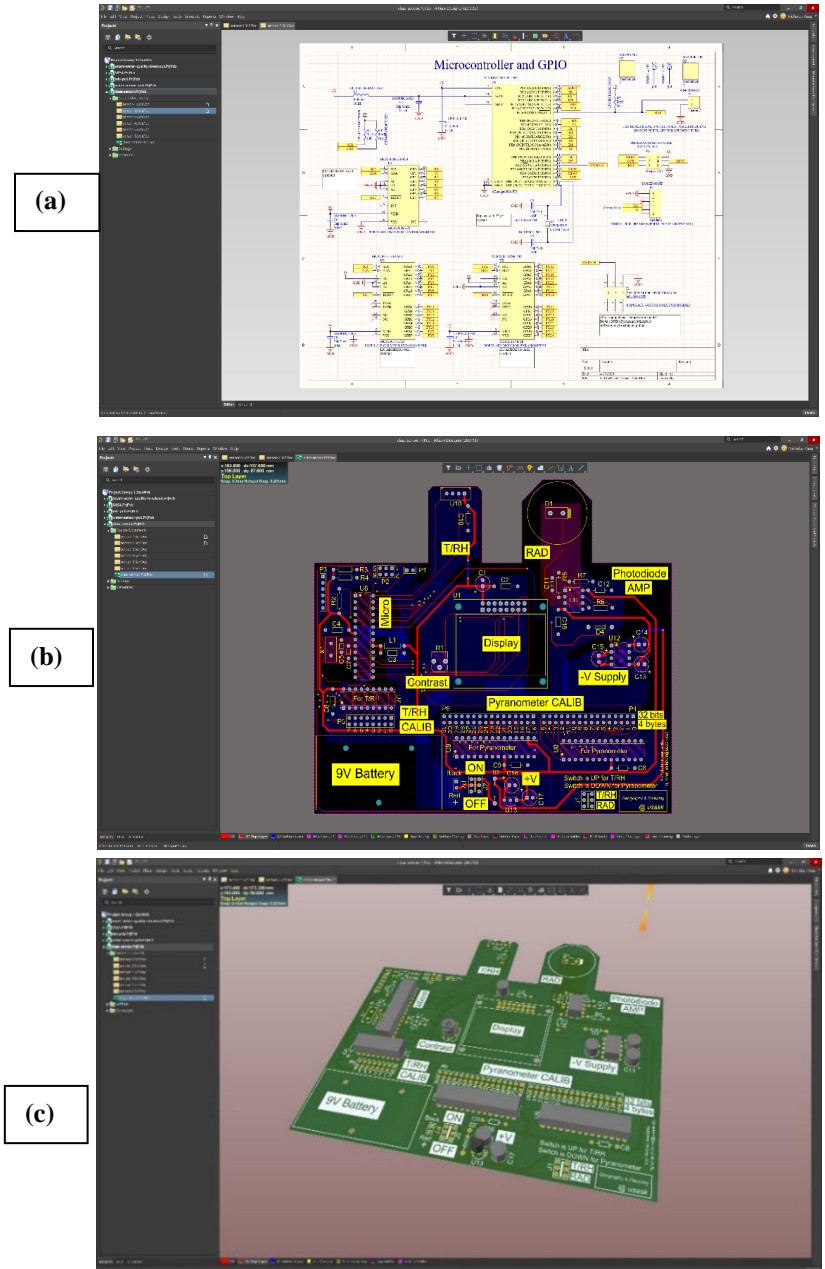

**Fig. 2.** (a) Electronic layout software showing a schematic page of the hybrid temperature, relative humidity, and solar radiation sensor. (b) Associated PCB with layers stacked in a similar fashion to a geographic information system (GIS). (c) 990 3D model of the PCB generated using the layout software.

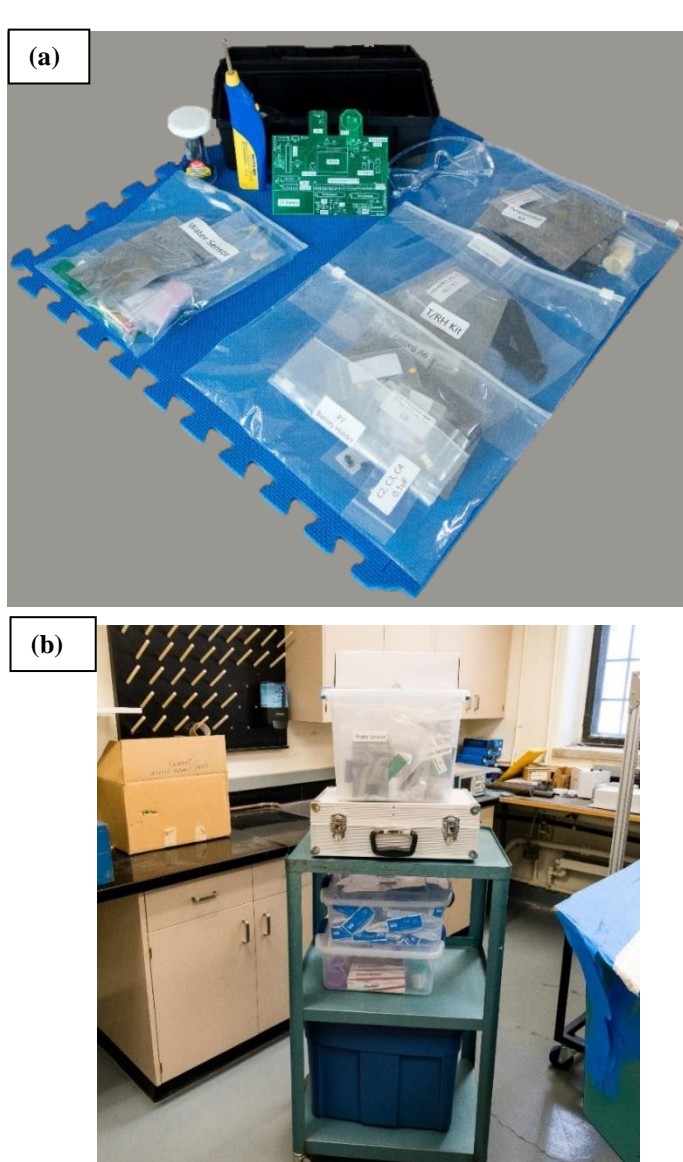

**Fig. 3**. (a) The circuit components were individually bagged and labelled for student assembly. Also visible in this image is
the battery-operated soldering iron used by students for soldering, a toolbox containing hand tools for assembly and a glue
bottle used to glue on the diffuser to the circuit board to cover the photodiode for creation of a pyranometer to measure
shortwave radiation from the sun. (b) Image showing containers used to transport the kits to a classroom.


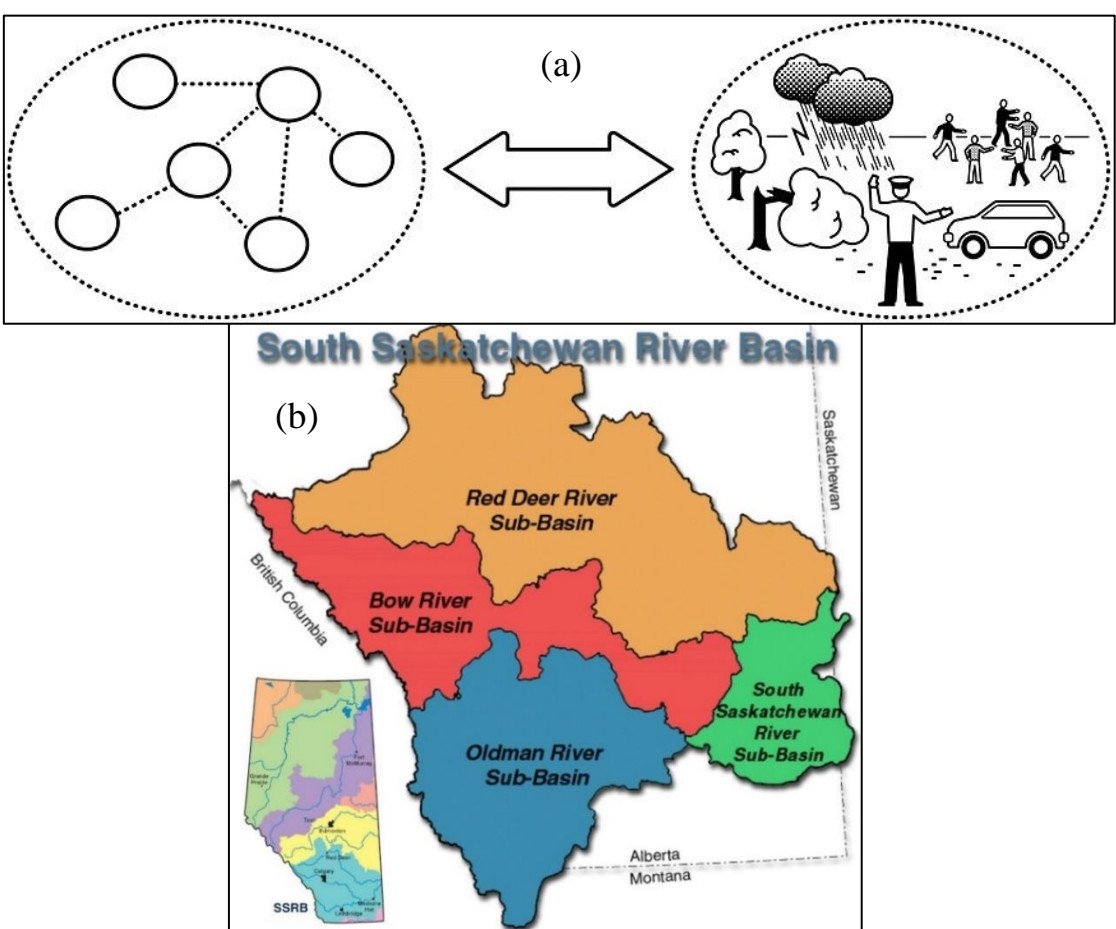

**Fig. 4**. (a) Demarcation of a system using control volumes and lines (Hieronymi, 2013). (b) Sub-basins used as an in-class example (Sauchyn et al., 2016).


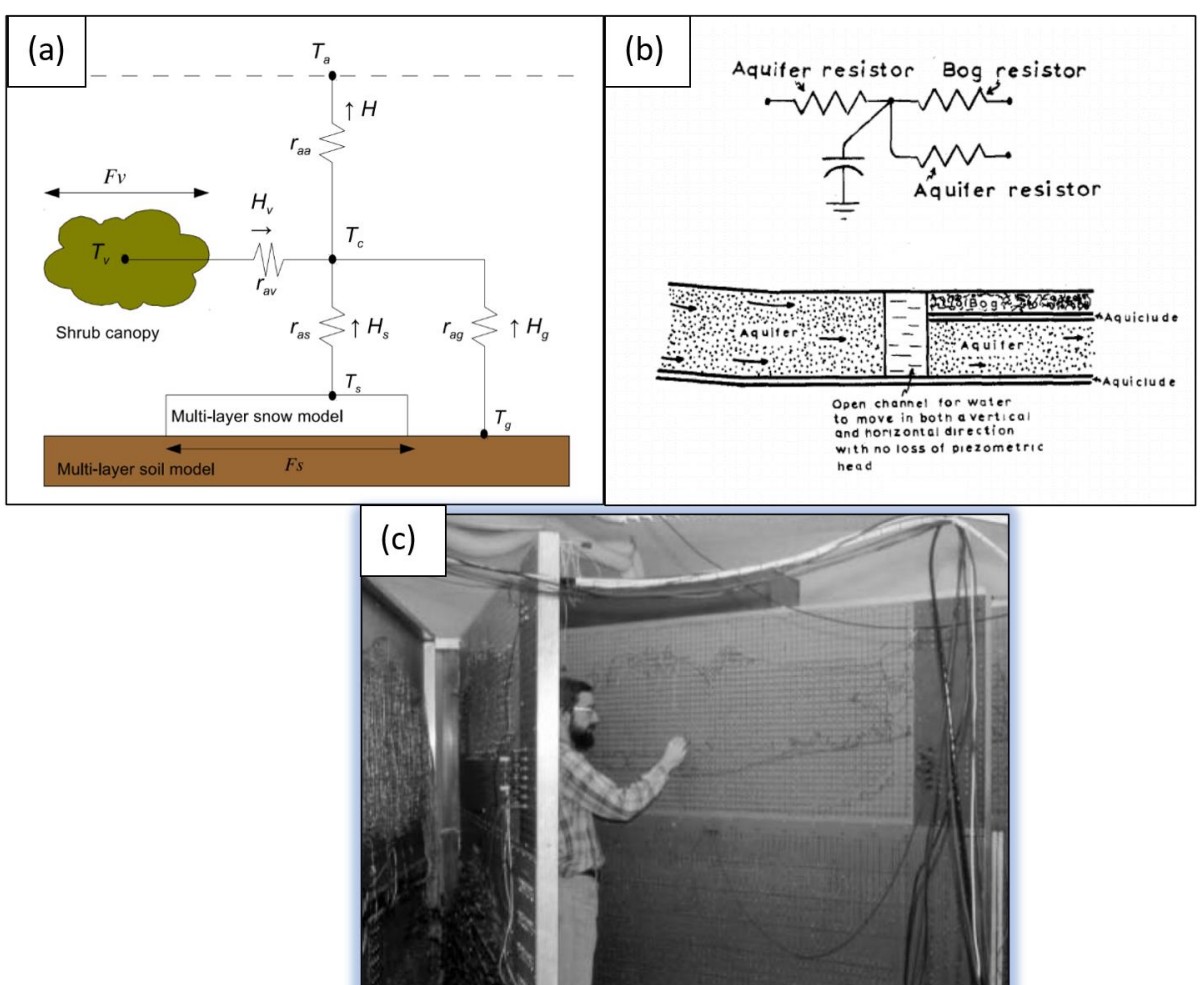

**Fig. 5**. (a) Example of an analog model used for modelling heat exchange between snow, soil, a shrub canopy and the atmosphere at a sub-Arctic tundra site (Ménard et al., 2014). For a complete explanation of terms, please see Ménard et al. (2014). Visible on this circuit schematic are resistors connected to form a circuit. (b) Analog model used to represent the flow of water in a peat bog (Sander, 1976). Visible on this circuit schematic are resistors and capacitors. (c) Picture showing a large analog circuit in the 1970s used to model groundwater (Reilly, 2004). Figures 2a-c are provided under license by John Wiley and Sons.


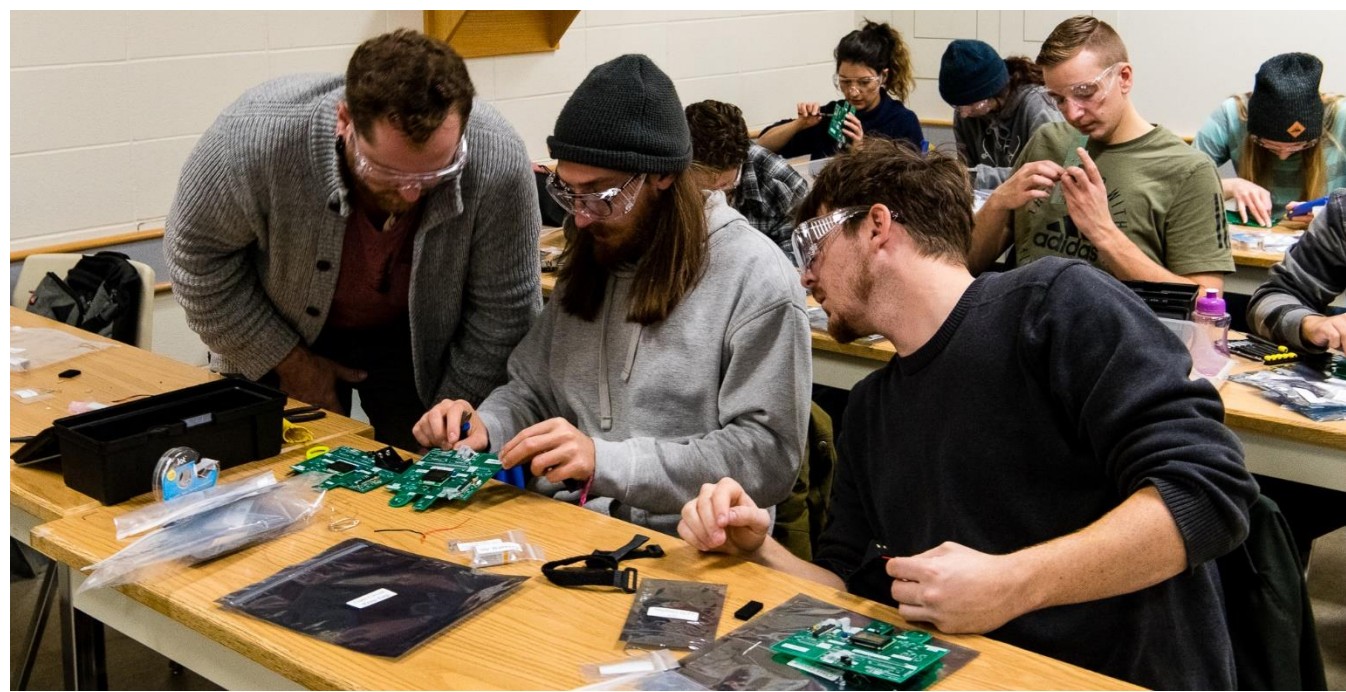

**Fig. 6**. Example images of circuit assembly in the class. The students are using battery-operated soldering irons to solder components to the PCB.




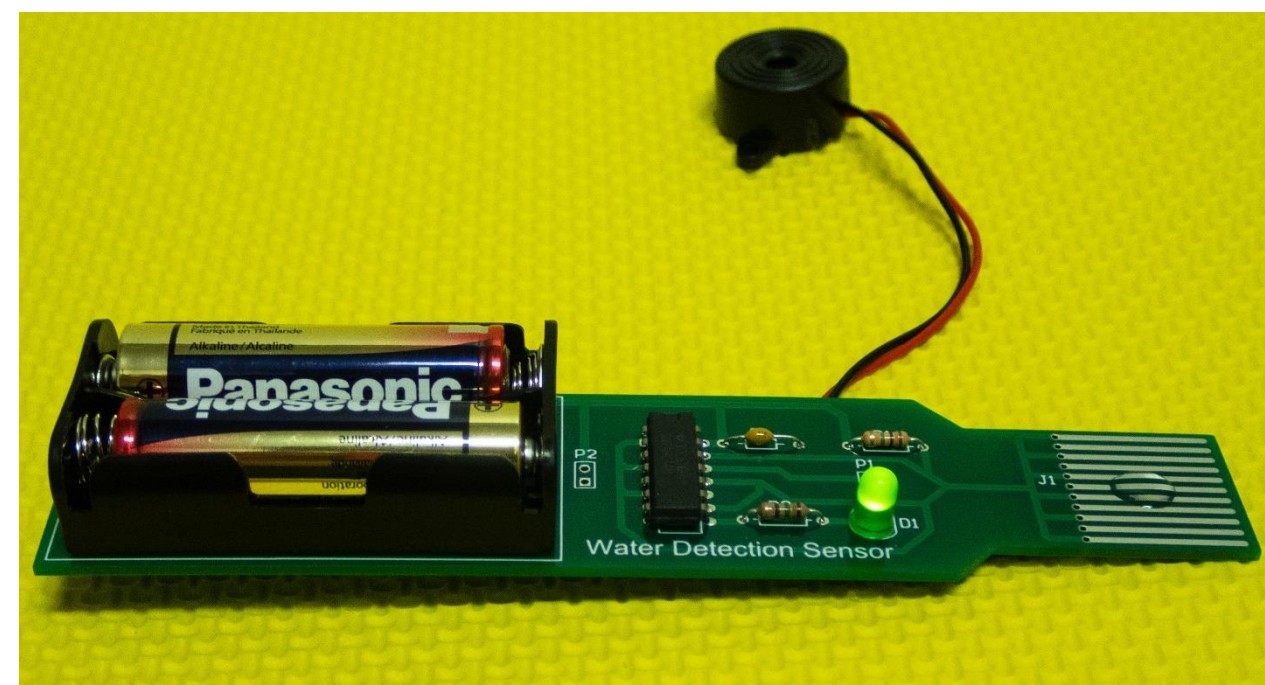

**Fig. 7**. Water detection PCB. A droplet of water turns on the light-emitting diode (LED) and an attached piezoelectric buzzer to provide visual and audible feedback.




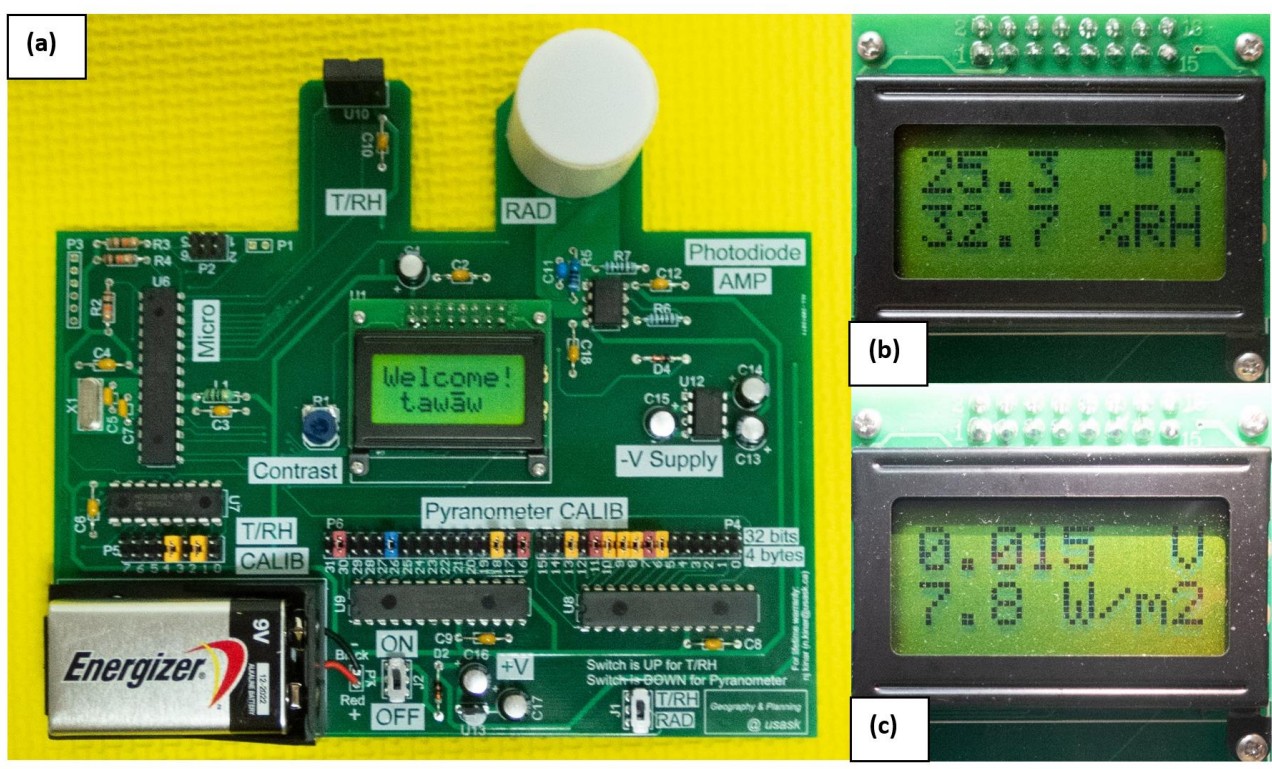

**Fig. 8.** (a) Hybrid temperature, relative humidity (RH) and solar radiation sensor. (b) LCD display showing calibrated temperature and relative humidity (RH) readout. (c) LCD display showing calibrated voltage output and radiation flux in $W\,m^{-2}$ . Calibration coefficients are set using removable plastic jumpers and the position of each calibration header is indicated by "CALIB" marked on the PCB silkscreen. The "T/RH" marking indicates the position of the temperature and RH sensor, whereas the "RAD" marking indicates the position of the pyranometer sensor comprised of a photodiode and an 1080 op-amp. The photodiode is covered with a piece of plastic pipe and a circular Teflon diffuser. This ensures that the photodiode has a cosine response to obtain a hemispherical measurement. Other markings indicate the positions of various PCB sub-systems, including a switch for setting the mode of circuit operation and a contrast adjustment potentiometer.



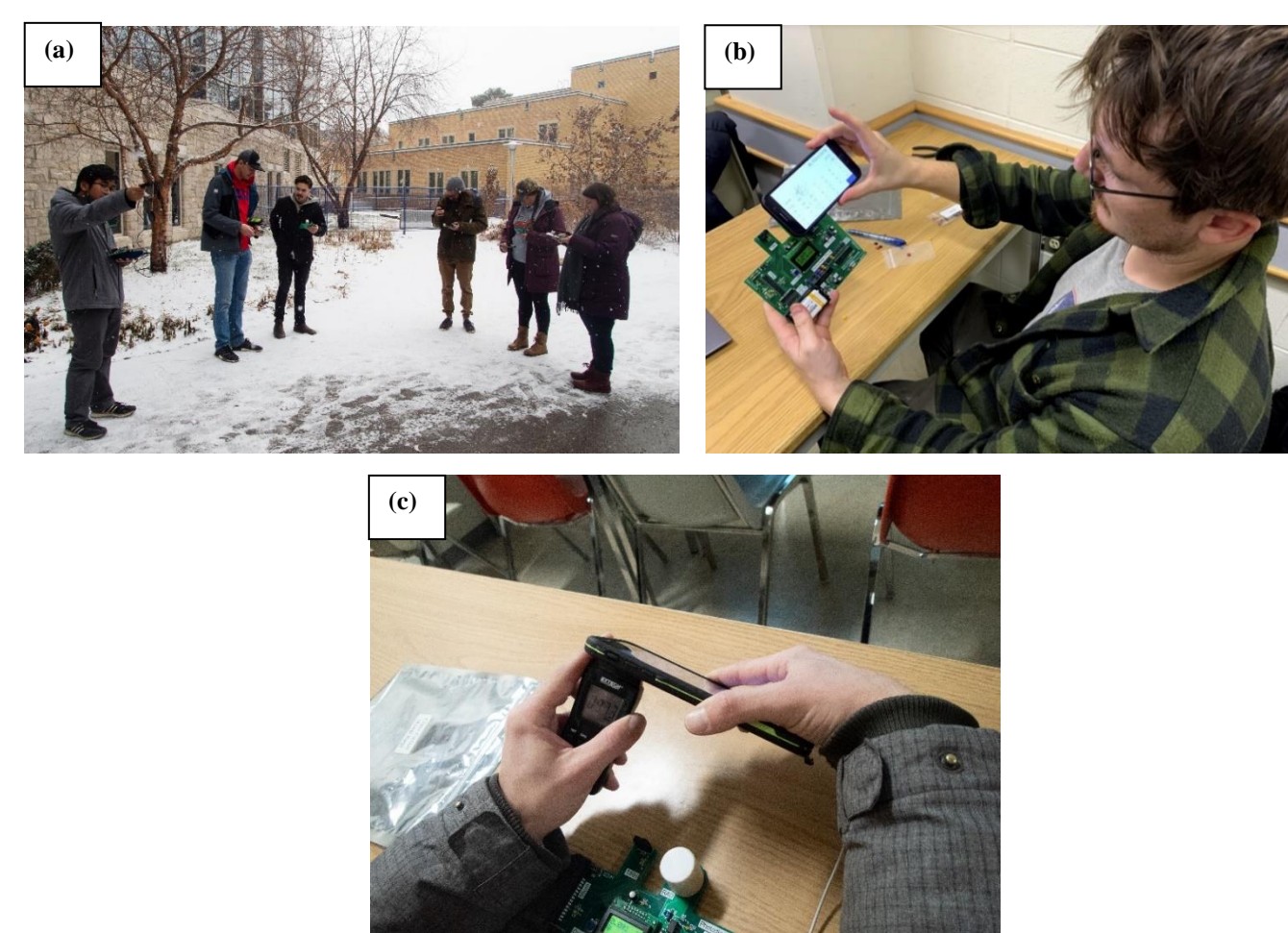





**Fig. 9.** Students engaged in calibration activities. (a) Outdoor calibration of temperature, RH and pyranometer devices using hand-held commercial sensors. Indoor calibration (b, c) with a smartphone flashlight and a hand-held solar radiation meter used as a standard for comparison.


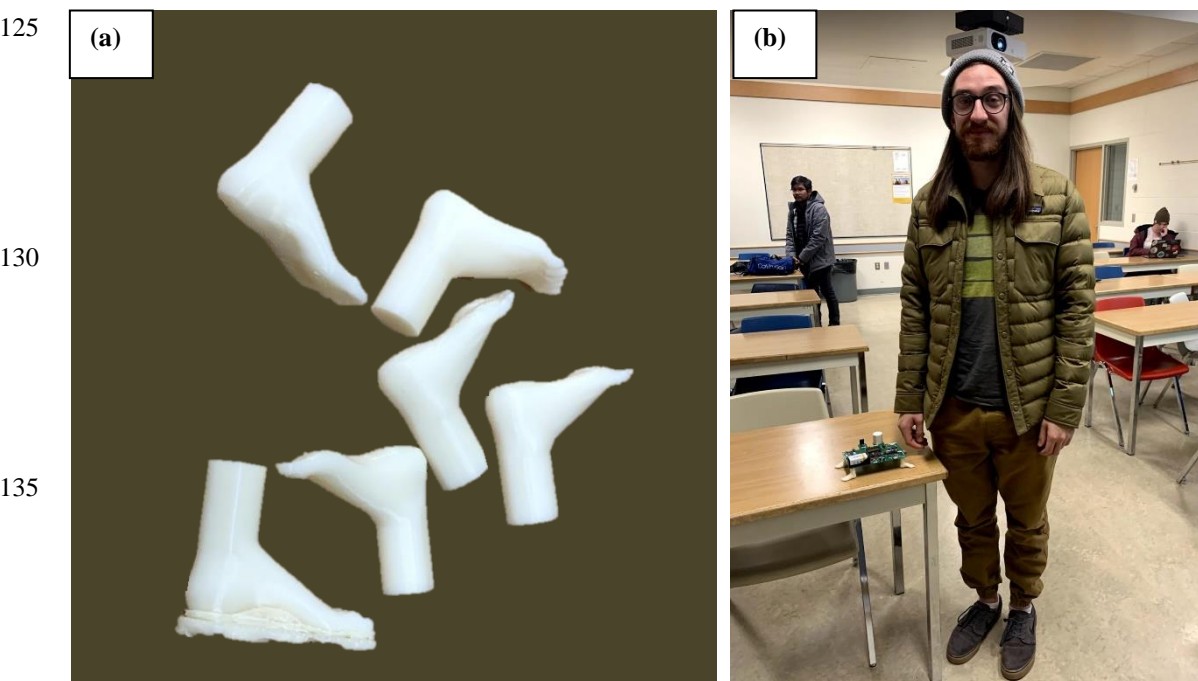

**Fig. 10**. (a) Example of 3D printed "feet" to be glued on the bottom of a PCB. The 3D printed "foot" nearest the bottom of the image has a 3D printed "raft" that has not been removed. The "raft" is placed on the 3D printer build platform to ensure that the deposited plastic sticks to a substrate. (b) Student next to a PCB exhibiting "feet" glued on the bottom.

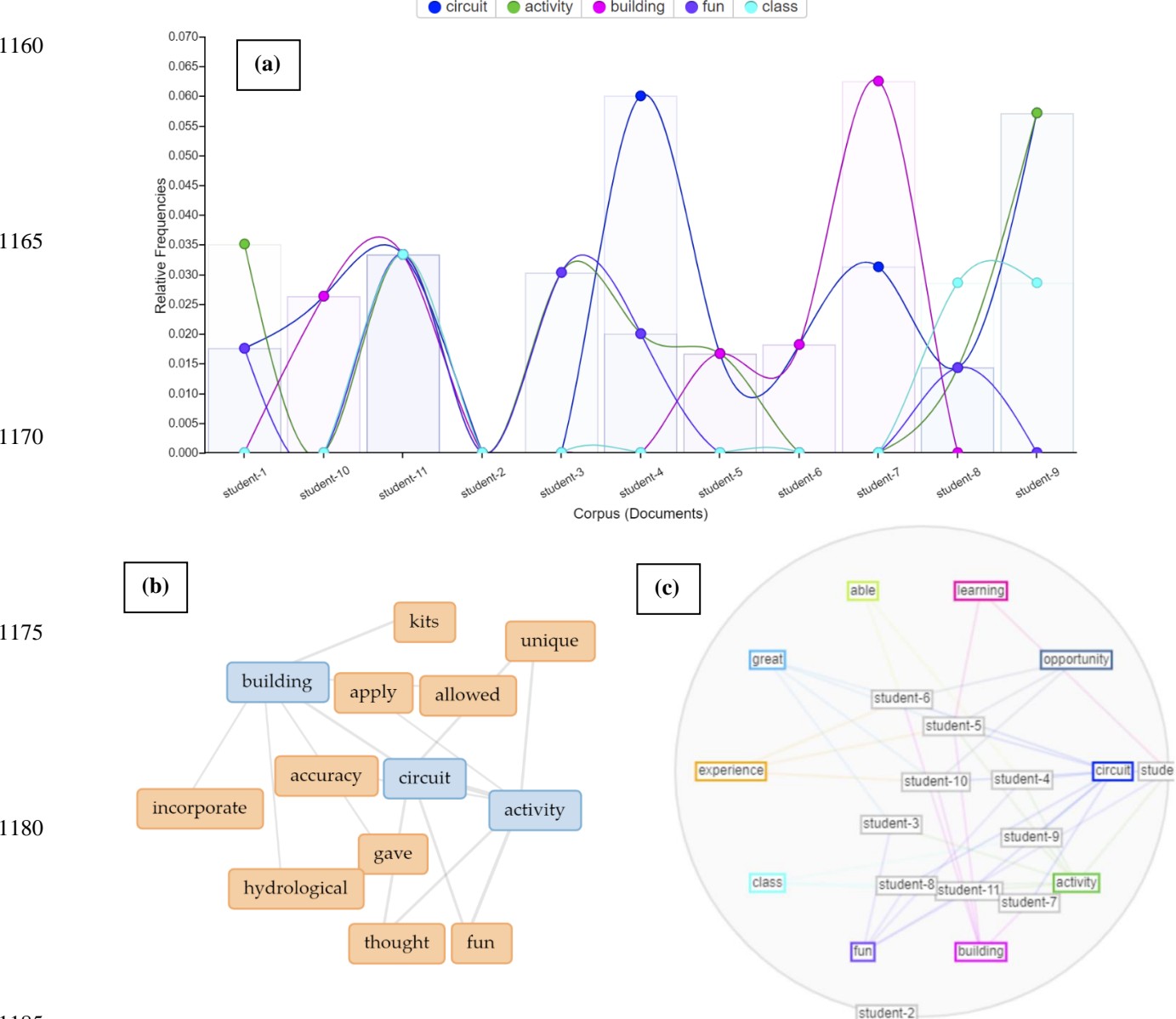

**Fig. 11**. Text analysis for the open-ended class feedback using the Voyant software. (a) Trends plot showing words that occur most often with respect to student responses. The "Corpus" in the Voyant software refers to the student responses. (b) Links plot showing relationships between words over all responses. (c) Mandala plot indicating relationships between student responses and words.

| Manufacturing Defect | Number of PCBs Affected |
|---|---|
| Soldering joints | 9 |
| Components placed with wrong orientation | 5 |
| Battery connections | 3 |
| Glue dripping on photodiode from diffuser | 4 |
| Battery holder breaking | 3 |
| Damaged PCB pads | 1 |

**Table 1**. Estimated list of defects associated with PCBs created by the students.

| PCB Configuration | Number of Students |
|---|---|
| Water detection circuit | 12 |
| Relative humidity (RH)/air temperature | 15 |
| Pyranometer | 0 |
| RH/Temperature/Pyranometer | 13 |
| **Total Students in Class** | 18 |

**Table 2**. Number of students with completed circuit configurations.

| Student Number | Response |
|---|---|
| 1 | I thought the circuit activity was very informative.  We spend our time learning concepts in other classes and not much time on the instruments being used. It's important to know the accuracy of the instrument.  This activity helped with the understanding and mechanics of the tools we use in the field. It was also lots of fun. |
| 2 | In the future maybe 1 soldering iron per 2 people maybe go + explain the calibration process more thoroughly |
| 3 | The board needs feet so it will sit level on a table.  The activity was fun and i learnt a lot.  The practice soldering was also great. The calibration was an interesting task. |
| 4 | I thought that the circuit activity was fun.  I have never built a circuit before & was a little nervous to, but this was a nice, easy introduction to it.  It's cool to be able to have your own circuit that measures actual values useful to you that you built yourself. |
| 5 | The circuit activity was a very unique and beneficial experience!  I have not had the opportunity to do something such as this in my previous studies, & I'm thankful for all the time Professor Kinar put into building these kits for us.  Learning about the process used to build circuits, even unrelated to hydrological processes, has been a great learning experience! |
| 6 | Providing the opportunity for students to build and calibrate their own circuit boards is such a great hands on experience!  Being able to be a part of the building process from A to Z really helped me to understand how these sensors work, and I will be able to use these skills in the future. |
| 7 | I liked how the building kits were organized and how every bag contained the necessary components I don't like how long the circuit building took overall, but it is how it is. |
| 8 | The circuit activity is a fun, educational, and hands-on way, of breaking up the class.  It keeps students interested and gave them something they can take home and show for their work in the course.  I also found that it encouraged attendance of the class. Professor Kinar, clearly a lot of work was put in on your behalf and |

| | |
|---|---|
| | explainations were very clear for the most part. Thank you! |
| 9 | The circuit activity was very unique. I found it very useful and |
| | I was more engaged in the class. The circuit activity did take multiple |
| | classes to complete but I still found it very enjoyable. |
| 10 | This exercise was a unique experience that I likely would not have had the opportunity to do |
| | if not a part of the Advanced Hydrology course. It was a great way to incorporate Hydrological |
| | Modelling and circuit building. |
| 11 | This exercise was very fun and gave us insight to circuit building. |
| | It allowed us to make correlations between what we learnt in class and apply it |
| | to an activity. |

**Table 3**. Student open-ended feedback responses transcribed as written by the students. The student number in the first column of the table was only used for analysis and is not intended to covey a ranking of responses. Moreover, the student number cannot be used to identify a student in the class. The responses in the table above are also available for digital download (Section 7).
