# Peer review of "Introducing Electronic Circuits and Hydrological Models to Postsecondary Physical Geography and Environmental Science Students: Systems Science, Circuit Theory, Construction and Calibration"

_Geoscience Communication, 2020_

## Editor Comment (EC1) · Stephanie Zihms (Editor) · 14 Jul 2020

This paper is very interesting as it shows that an enquiry into how or why teaching works can really inform future practice in the class room. It also shows how we can engage more in scholarship around learning and teaching. Lessons learned from activities like this in the class room would also translate well into science communication and can help design those as well.

[Figure]

**[GCD](...)**

---

## Author Comment (AC1) · 14 Jul 2020

Thanks for this comment. I agree that there is a need for geoscience communication in the classroom. I also strongly believe that to effectively facilitate communication, a classroom activity should be set up for replication, so all design files and a bill of materials should be provided under a permissive open source license for re-use, re-mixing and sharing.

---

## Referee Comment (RC1) · Anonymous Referee #1 · 17 Aug 2020

General Comments Overall, this is a very interesting paper that explains the planning and execution of a circuit building and calibration activity in an advanced undergraduate hydrology class. I think the information that is presented will be of interest to other educators teaching hydrology and climatology classes. The author does a good job of providing details on all aspects of the activity and how the activity was received by the students. I think the paper would benefit from clearly identifying what the questions and/or objectives of the study are and restructuring the methods and results to

separate out conceptual and technical information, as well as ensuring that results are not presented in the Methods section. For example, the information on different types of systems in section 2.1 Background could be integrated into the Introduction section to provide a rationale for the study. A lot of this information seems out of place in the Materials and Methods section. As well, throughout the paper, very short paragraphs (2-3 sentences) are used (e.g. L248-57). While I can see the benefit of this in some places, I think the overall flow of the paper is hampered with this approach. For example, in section 2.3, the focus of the paragraphs jump around quite a bit (technical information about the calibrations, student feedback). Overall, I recommend the author try to integrate some of these shorter paragraphs together under common themes. In the results and discussion section, the author should try to make clear links back to the research questions/objectives, i.e., try to make the different sections of the paper mirror each other.

Specific Comments L53: I recommend changing 'will eventually' to 'may'. L67: Unclear what 'threshold concepts' is referring to in this sentence. L76-83: I recommend trying to tighten up this paragraph a bit as there is some repetition. For example, 'Before circuits are constructed. . .' and 'Prior to circuit construction. . .'. L110-3: The mention of HRUs, CRHM and SWAT seem unnecessary here. They do not further the explanation of systems in hydrology in any way. I recommend removing these or rewording to make it clear why this information is useful for understanding hydrological systems. L210-4: This paragraph seems out of place here since below the actual details of the activity are explained. L225: I recommend presenting the methods sequentially. This paragraph describes what happened before the activity started, so present it first. L231-2: Was this based on written feedback? Or just the verbal feedback? L258-266 and L280-4: This seems like information that should be in the Results and/or Discussion section. L297-305: Section 2.4 is really a result of the study. I recommend moving it into section 3. L308-24: Section 2.5 seems more suitable for a Discussion section. L412-414: Instead of using the trends plot, I recommend providing quotes from the students. Otherwise the reader is left to guess the context for each of these words. L425-35: I'm

not sure that the overall course survey feedback is relevant here as it's really the circuit building/calibrating activity that is being discussed. L456-7: While good to hear, I don't think that feedback on the instructor for a single course is relevant here. I recommend that the author just focus on student feedback related to the circuit activities. L562: I'm not sure that the inclusion of information on the 3D watersheds is helpful in this paper since the focus is really on the circuit building. L574-620: I recommend trying to shorten the Conclusions. There is information in here that could be placed back in the Discussion (e.g., recommendations for future classes). Figure 3: I suggest choosing just one of these photos.

---

## Author Comment (AC2) · 17 Aug 2020

Thank you very much for your comments. I will incorporate these comments into the revised draft and restructure as indicated.

---

## Referee Comment (RC2) · Nilay Dogulu (Referee) · 29 Sep 2020

It was a great pleasure to review this manuscript. However, I agree with the Editor that there is an overwhelming focus on the technical details. There must be an equal focus, e.g. on aspects related to hydrology education and overall (methodological) design of the activity, to balance out such skewness. In my report, there are some suggestions for mainly the Introduction and Conclusion parts to address this issue.

There is a good potential in this manuscript, and I look forward to the revised version.

Please also note the supplement to this comment:
https://gc.copernicus.org/preprints/gc-2020-30/gc-2020-30-RC2-supplement.pdf

[Figure]

**Supplement:**

Introducing Electronic Circuits and Hydrological Models to Postsecondary Physical Geography and Environmental Science Students: Systems Science, Circuit Theory, Construction and Calibration – by *Nicholas J. Kinar*

**General Comments**

The manuscript by *N. J. Kinar* shares the experiences gained from a class activity held in a fourth year Hydrology class offered at the University of Saskatchewan. The activity is aimed at synthesizing hydrological process knowledge from a systems approach perspective. Inputting the collected data by the low-cost instrumentation using electronic circuits design, students were asked to build a hydrological model for gaining insights into mathematical modelling in environmental sciences.

First and foremost, I would like to thank N. J. Kinar for his motivation to invest in designing innovative activities for advancing hydrology teaching in line with growing technology opportunities. I believe that this work sets a good example on how small efforts like designing such class activities during early years of hydrology education (i.e. postgraduate level) can prove valuable in shaping the future minds that are able to connect the dots in trying to solve today's (and future's) increasingly complex and growing environmental problems in the most efficient and plausible manner.

In general, the manuscript conveys a good quality content and is well-written. There are, however, a number of opportunities to make it more successful. Repetition of information is the biggest issue that prevails throughout the manuscript. Paragraphs with only a few sentences are very common (it is not disturbing me as it makes the reading easier), in some parts not really necessary though. Merging (e.g. last two paragraphs in Sec 3.3) or creating a bullet point list (e.g. "the students found that" in P9) can be considered.

Please see below a summary of my observations that I would like draw the attention of the author.

- ♥ **The Title** reads fine. It is long but comprehensively highlights the paper's scope to the potential targeted audience. **Abstract** and **Short Summary** is written in a concise manner and include key information about the paper. **Supplementary Material** provides the required background knowledge on the electronics and some practical considerations, as well as explains in detail the three circuits (Water Detection Circuit, Relative Humidity (RH)/Air Temperature Circuit, Pyranometer Circuit) whose descriptive schematics are available via figshare for downloading.

- ♥ **Introduction:** It can definitely benefit from drawing a wider picture for setting the background in relation to fundamental links to practical considerations. For details, please see my comment on **P2 L33** under "Specific Comments".

- ♥ **Research Question & Objectives:** Not mentioned at all. Please add a paragraph stating the objective of this manuscript. Obviously, one objective is to share experiences (i.e. challenges, tips and resources), and another is to highlight the benefits of integrating simple hands-on exercises (using simple technology) into teaching curriculum to enhance active learning in geo- and environmental sciences, particularly in hydrology education.

- ♥ **Materials and Methods (Section 2):** I don't think that this title is appropriate. This section is overloaded and mixed, needs to be revised. A new outline with properly structured sections (subsections) which enable the reader to follow the manuscript with much less confusion is a must.

**Geoscience Communication / GC-2020-30**

Introducing Electronic Circuits and Hydrological Models to Postsecondary Physical Geography and Environmental Science Students: Systems Science, Circuit Theory, Construction and Calibration – by *Nicholas J. Kinar*

- ♥ **Methodology:** Open a new section please. A workflow diagram would be really helpful here. It should depict different stages involved in every main phase of this class activity (i.e. pre-, during- and after). A brief summary should be added explaining which methods/tools/etc. are considered/used in each before describing each step in detail in subsections.

- ♥ **Results (Section 3):** The subsections "3.1 Manufacturing Defects" and "3.2 Choice of Configuration" don't seem to fit well in here. These shall be moved to a new section titled "Electronic Circuits". "3.3 Open-Ended Feedback" and "SLEQ Feedback" form the core of this part. Instead of "Results", this section can be titled such that it tells the reader now is time to read the feedback by the course participants. Section 4 "Discussion" should be combined in this section too.

- ♥ **Outline:** Thus, my new outline suggestion is:

  1. *Introduction*
  2. *Methodology*
     Be more specific about stages of the class activity (i.e. pre-, during- and after).
     a. *Course (activity) proposal and acceptance*
     b. *Technical setup material purchase*
     c. *Preparation of handouts, guidance material for students*
     d. *Implementation of the activity*
     e. *FAQ*
     f. *…*
     g. *Design, Analysis and Collection of Written Feedback*
  3. *Systems Approach and Hydrological Modelling*
     Try to create two or three subsections on the text provided in "2.1 Background". Be careful not to overlap with the Introduction part. In other words, avoid repetition.
  4. *Electronic Circuits*
     a. *Theory and Construction*
     b. *Classroom Application*
              It is too long as a whole, but some paragraphs are very short. Try to create sub-subsections to clearly present various aspects presented in Sec 2.3.
              Manufacturing defects and choice of configuration shall be mentioned here.
     c. *Example Applications for Electronic Circuits*
     d. *Modelling with Electronic Circuit Data*
  5. *Feedback*
     a. *Open-Ended Feedback*
     b. *SLEQ Feedback*
     c. *Feedback Loops (i.e. Sec 4 Discussion)*
  6. *Conclusion*

♥ **Conclusion:** Overall not so good. I especially liked the paragraphs on P19 (L598-620). However, these are essentially the summary of feedback from the students. My suggestion is to relocate this into (new) Section 5 "Feedback". P18 L587-591 should be in the Methodology. Conclusion can be concise. It is more important to adopt a more general perspective into the scope and purpose of the study rather than sharing (again) various details of the activity. See also my comment on the Introduction. This (conclusion) is the part where the author can take the floor to convince the reader that hydrology education needs improvements which incorporate innovative class activities into teaching strategies adopted by the lecturers (and invite more lecturers to embrace this strategy). A discussion on the importance of teaching aspect of a "professorship" position can be given too. (e.g. a call for universities to accommodate such needs by distinguishing explicitly between "lecturer" and "researcher" positions. And that courses should be designed by a joint team so that well-defined teaching strategies supported with innovative and fitting class activities can be formulated addressing the needs of students and in line with emerging topics & technologies in hydrology.)

**Specific comments**

**P2 L33.** The connection to the previous paragraph can be strengthened in the following manner:

First, set the scene by mentioning that geography and environmental science teaching (recommended to) involve(s) different components, i.e.: introduction of conceptual theory, examples of how theories are implemented in real life applications and relevance to address societal challenges (addressed by the United Nations Sustainable Development Goals, UN SDGs), class activities to consolidate information to knowledge for achieving efficient learning of students. Then it is safe to bring forward the latter (class activities) and explain further. Paragraph 1 (in the current version of the manuscript) fits into here. Also, it will be good to include some comments on how such activities encourage development of a diverse set of skills, as well as if (by training future's professionals) and how such skills contribute to Multi-, Inter-, Trans- disciplinary research and practice.

Secondly, provide a brief overview on the progress and needs of hydrology education (see the list of suggested reading). Give some examples of class activities from the hydrology literature.

Thirdly, highlight the importance of data for science and teaching: The essential ingredient for geography and environmental sciences is data. Data collection is the backbone of advances in geosciences (thus teaching). In hydrological sciences education, water data (discharge and/or water level) collection is traditionally covered in many hydrology courses, however in rather a traditional way (e.g. in terms of measurement techniques) and often with a limited scope within a short duration. Maybe merge Paragraph 3 into this section. I recommend citing Tauro et al. (2017) to give an overview on the efforts to advance hydrological sciences through innovative measurement techniques. Also, the need for such efforts in view of not only providing support to financially disadvantaged countries but also harnessing the power of harmony in unity (e.g. WMO HydroHub).

Lastly, explain how electronic circuits are used for data collection purposes in hydrology and meteorology. Paragraph 2 fits into here, followed by Paragraph 4, 5 and 6.

**P2 L59-63.** How is the open source electronics movement linked to the Open & FAIR data? Briefly mention.

**P3 L70.** Before Paragraph 7, please state the research objective (and associated questions). L74-75 are the objectives of the course taught, not the manuscript's.

**P3 L76-93.** Paragraph 8-9-10 should be moved to Section 2 "Materials and Methods" (Sec 2.3 in particular) as they describe several aspects related to pre-, during- and after the class activity (i.e. electronic circuit construction).

**P18 L587-591.** Better if moved to Section 2 "Materials and Methods" (before subsections, a general summary can be given supported with a diagram showing all the stages).

**P19 (Code and Data Availability).** Thanks for your dedication to Open Science and FAIR Data.

**P20 (Author contribution).** Is there really a need to specify all tasks individually? Maybe one sentence is enough? E.g. All tasks (conceptualization, methodology, software, ….) are contributed by N. K.

**P20 (Acknowledgements).** Thanks for being so elaborate and honest.

**P21 (References).** Quite a comprehensive list, impressive. Properly cited, incorporated well into the text.

*Minor Edits*

**P1 L14.** "systems science, models in hydrology, and calibration" > "systems science, modelling in hydrology, and model calibration"

**P3 L71.** Abbreviation (USask) can be used for the "University of Saskatchewan" in the remaining part.

**P4 L115&122, P5 L160.** Students in the Advanced Hydrology > Just say "students".

**P7 L213&L222, P12 L430, P17 L529…** The name of the course "Advanced Hydrology class" doesn't need to be indicated every single time.

**P13-15 L405-470.** The texts quoted should be written in Italic.

**P18 L578.** Add in parenthesis the English translation of "tawāw".

**P18 L581.** allowed for > allowed

**P18 L584.** Instead of "hydrology", use "technology".

**P18 L586.** Reformulate the sentence, maybe better as follows: "This paper provides explanations on Circuit theory and relevant concepts for classroom implementation and replication."

**References (used in this review)**

Tauro, F., Selker, J., Van De Giesen, N., Abrate, T., Uijlenhoet, R., Porfiri, M., ... & Ciraolo, G. (2018). Measurements and Observations in the XXI century (MOXXI): innovation and multi-disciplinarity to sense the hydrological cycle. Hydrological Sciences Journal, 63(2), 169-196. https://doi.org/10.1080/02626667.2017.1420191

HydroHub by the World Meteorological Organization https://hydrohub.wmo.int/en/home

**Suggested Reading**

Kingston, D. G., Eastwood, W. J., Jones, P. I., Johnson, R., Marshall, S., Hannah, D. M., & Seibert, J. (2012). Experiences of using mobile technologies and virtual field tours in Physical Geography: implications for hydrology education. Hydrology & Earth System Sciences, 16(5). https://doi.org/10.5194/hess-16-1281-2012

Ruddell, B. L., & Wagener, T. (2015). Grand challenges for hydrology education in the 21st century. Journal of Hydrologic Engineering, 20(1), A4014001. http://dx.doi.org/10.1061/(ASCE)HE.1943-5584.0000956

Seibert, J., Uhlenbrook, S., & Wagener, T. (2013). Preface" Hydrology education in a changing world". Hydrology and Earth System Sciences, 17(4), 1393-1399. https://doi.org/10.5194/hess-17-1393-2013

Van Loon, A. F. (2019). Learning by doing: enhancing hydrology lectures with individual fieldwork projects. Journal of Geography in Higher Education, 43(2), 155-180. https://doi.org/10.1080/03098265.2019.1599330

Wagener, T., Weiler, M., McGlynn, B., Gooseff, M., Meixner, T., Marshall, L., ... & McHale, M. (2007). Taking the pulse of hydrology education. Hydrological Processes: An International Journal, 21(13), 1789-1792. https://doi.org/10.1002/hyp.6766

---

## Author Comment (AC3) · 30 Sep 2020

Thank you for these insightful comments; I will restructure the document according to the suggestions. I appreciate the revised outline and the time spent reviewing this paper.

---

## Author Response (AR1)

**Author Responses to Referees**

The referee response is given in italic purple text, whereas the author responses as a list of changes are given in non-italic text under each referee response. In the edited paper, major additions to the text are indicated in a blue color and line or page numbers are also provided in the responses below. Revisions have been made to the paper structure along with an associated editing of content to address the referee responses. A version of the manuscript showing differences is also attached to this response file.

**Anonymous Referee #1**

*Overall, this is a very interesting paper that explains the planning and execution of a circuit building and calibration activity in an advanced undergraduate hydrology class. I think the information that is presented will be of interest to other educators teaching hydrology and climatology classes. The author does a good job of providing details on all aspects of the activity and how the activity was received by the students.*

Thank you for these comments and for your recommendations.

*I think the paper would benefit from clearly identifying what the questions and/or objectives of the study are and restructuring the methods and results to separate out conceptual and technical information, as well as ensuring that results are not presented in the Methods section. For example, the information on different types of systems in section 2.1 Background could be integrated into the Introduction section to provide a rationale for the study. A lot of this information seems out of place in the Materials and Methods section.*

- The objectives of the study are now listed in point form in the Introduction section of the paper (pg. 5).
- The background information previously in Section 2.1 has now been moved to the Introduction section (Section 1).
- The information in the materials and methods section has been restructured to separate conceptual and technical information.

*As well, throughout the paper, very short paragraphs (2-3 sentences) are used (e.g. L248-57). While I can see the benefit of this in some places, I think the overall flow of the paper is hampered with this approach. For example, in section 2.3, the focus of the paragraphs jump around quite a bit (technical information about the calibrations, student feedback). Overall, I recommend the author try to integrate some of these shorter paragraphs together under common themes.*

- The shorter paragraphs have been combined into larger paragraphs for presentation throughout the paper and some paragraphs have been eliminated or restructured, as necessary. Lines L248-57 in the original draft have been removed and replaced in an associated discussion (see the second paragraph of Section 2.2).
- The manuscript has now been organized by common themes.
- This restructuring approach has also been implemented with respect to the suggestions given by Reviewer #2. Please see the response to Reviewer #2 for further information related to the restructuring changes used for this paper.

*In the results and discussion section, the author should try to make clear links back to the research questions/objectives, i.e., try to make the different sections of the paper mirror each other.*

➢ The results and discussion section (now Sections 4-5) indicate the objectives of the paper identified on pg. 5.

*Specific Comments L53: I recommend changing 'will eventually' to 'may'.*

➢ The wording has been changed as recommended (line 127).

*L67: Unclear what 'threshold concepts' is referring to in this sentence.*

➢ The sentence has been re-written for clarity (line 151-153).

*L76-83: I recommend trying to tighten up this paragraph a bit as there is some repetition. For example, 'Before circuits are constructed: : :' and 'Prior to circuit construction: : :'.*

➢ The repetition in this sentence has now been removed as recommended.  Please see the first paragraph of Section 2.4.

*The mention of HRUs, CRHM and SWAT seem unnecessary here. They do not further the explanation of systems in hydrology in any way. I recommend removing these or rewording to make it clear why this information is useful for understanding hydrological systems.*

➢ For clarity and presentation, references to these models have been removed from the paper (Section 3) and Figure 4 updated.

*L210-4: This paragraph seems out of place here since below the actual details of the activity are explained.*

➢ The paragraph has been removed from the paper for clarity and to reduce repetition.

*L225: I recommend presenting the methods sequentially. This paragraph describes what happened before the activity started, so present it first.*

➢ This paragraph has now been written and the information moved to Section 2.4.

*L231-2: Was this based on written feedback? Or just the verbal feedback?*

➢ This is a verbal response during the class activity.  Please see Section 4.2 for an updated version of these lines.  All verbal responses during the class circuit activity are given in Section 4.2, whereas feedback collected after the class circuit activity is discussed in Section 5.

*L258-266 and L280-4: This seems like information that should be in the Results and/or Discussion section.*

➢ This information has been moved to Section 5.1.

*L297-305: Section 2.4 is really a result of the study. I recommend moving it into section 3.*

➢ The information in Section 2.4 has now been moved to Section 4.3 of the updated paper and paragraphs combined for presentation.

*L308-24: Section 2.5 seems more suitable for a Discussion section.*

➢ The information in Section 2.5 has now been moved to Section 4.4.

*L412-414: Instead of using the trends plot, I recommend providing quotes from the students. Otherwise the reader is left to guess the context for each of these words.*

➢ Quotes from the students are now provided in Table 3 and the quotes are analyzed by a discussion (pg. 17).
➢ The trends plot allows for five re-occurring words to be quantitatively identified in the text and the associated discussion is supplemented by Table 3.
➢ The student responses were checked before Table 3 was created.  A transcribing error for Student #7 was fixed before Table 3 was inserted into the document and Figure 11 was updated. The update of Figure 11 does not change the analysis presented in the text of this paper or the analysis given on pp. 16-17.  All quotes from the students are also available for download (Section 7).

*L425-35: I'm not sure that the overall course survey feedback is relevant here as it's really the circuit building/calibrating activity that is being discussed.*

➢ For clarity, the overall course survey feedback discussion was removed from the paper.

*L456-7: While good to hear, I don't think that feedback on the instructor for a single course is relevant here. I recommend that the author just focus on student feedback related to the circuit activities.*

➢ The instructor feedback has been removed from the paper so that the paper is focused on student feedback related to circuit activities.

*L562: I'm not sure that the inclusion of information on the 3D watersheds is helpful in this paper since the focus is really on the circuit building.*

➢ The information on 3D printed watersheds has now been removed from the paper to better focus the paper on the student activity.  Figure 4 in the revised paper has also been accordingly updated along with the Abstract.

*L574-620: I recommend trying to shorten the Conclusions. There is information in here that could be placed back in the Discussion (e.g., recommendations for future classes).*

The Conclusion section has been shortened and some information (recommendations for future classes) has been placed in the discussion sections of the paper.  Please see Section 6 and the end of Section 5.1.

*Figure 3: I suggest choosing just one of these photos.*

One of these photos of the class activity has been retained in the edited version of the paper (please see Figure 6 in the edited version of the document).

**Referee #2 – Nilay Dogulu**

*It was a great pleasure to review this manuscript. However, I agree with the Editor that there is an overwhelming focus on the technical details. There must be an equal focus, e.g. on aspects related to hydrology education and overall (methodological) design of the activity, to balance out such skewness. In my report, there are some suggestions for mainly the Introduction and Conclusion parts to address this issue. There is a good potential in this manuscript, and I look forward to the revised version.*

Nilay, thank you for your comments and recommendations. I have addressed your suggestions and a response to each suggestion is given below. The manuscript has been restructured as per the suggested outline.

*The manuscript by N. J. Kinar shares the experiences gained from a class activity held in a fourth year Hydrology class offered at the University of Saskatchewan. The activity is aimed at synthesizing hydrological process knowledge from a systems approach perspective. Inputting the collected data by the low-cost instrumentation using electronic circuits design, students were asked to build a hydrological model for gaining insights into mathematical modelling in environmental sciences.*

*First and foremost, I would like to thank N. J. Kinar for his motivation to invest in designing innovative activities for advancing hydrology teaching in line with growing technology opportunities. I believe that this work sets a good example on how small efforts like designing such class activities during early years of hydrology education (i.e. postgraduate level) can prove valuable in shaping the future minds that are able to connect the dots in trying to solve today's (and future's) increasingly complex and growing environmental problems in the most efficient and plausible manner.*

The suggestions that you have given help to support and indicate the arguments presented in this paper.

*In general, the manuscript conveys a good quality content and is well-written. There are, however, a number of opportunities to make it more successful. Repetition of information is the biggest issue that prevails throughout the manuscript. Paragraphs with only a few sentences are very common (it is not disturbing me as it makes the reading easier), in some parts not really necessary though. Merging (e.g. last two paragraphs in Sec 3.3) or creating a bullet point list (e.g. "the students found that" in P9) can be considered.*

- ➢ The repetition of information has been minimized throughout the document.
- ➢ The last two paragraphs in Section 3.3 of the first draft have been merged (pp. 16-17).
- ➢ A bullet-point list starting with "the students found that" has been added to the paper (pg. 14).

*The Title reads fine. It is long but comprehensively highlights the paper's scope to the potential targeted audience. Abstract and Short Summary is written in a concise manner and include key information about the paper. Supplementary Material provides the required background knowledge on the electronics and some practical considerations, as well as explains in detail the three circuits (Water Detection Circuit, Relative Humidity (RH)/Air Temperature Circuit, Pyranometer Circuit) whose descriptive schematics are available via figshare for downloading.*

➢ Thank you for these comments.

*Introduction: It can definitely benefit from drawing a wider picture for setting the background in relation to fundamental links to practical considerations. For details, please see my comment on P2 L33 under "Specific Comments".*

➢ The detailed comment has now been implemented in the revised version of the paper as per the blue text given in the revised document Section 1.

*Research Question & Objectives: Not mentioned at all. Please add a paragraph stating the objective of this manuscript. Obviously, one objective is to share experiences (i.e. challenges, tips and resources), and another is to highlight the benefits of integrating simple hands-on exercises (using simple technology) into teaching curriculum to enhance active learning in geo- and environmental sciences, particularly in hydrology education.*

➢ These objectives have now been added into this manuscript in the Introduction section of the manuscript (Section 1, bottom of pg. 5).

*Materials and Methods (Section 2): I don't think that this title is appropriate. This section is overloaded and mixed, needs to be revised. A new outline with properly structured sections (subsections) which enable the reader to follow the manuscript with much less confusion is a must.*

*Methodology: Open a new section please. A workflow diagram would be really helpful here. It should depict different stages involved in every main phase of this class activity (i.e. pre-, during- and after). A brief summary should be added explaining which methods/tools/etc. are considered/used in each before describing each step in detail in subsections.*

*Results (Section 3): The subsections "3.1 Manufacturing Defects" and "3.2 Choice of Configuration" don't seem to fit well in here. These shall be moved to a new section titled "Electronic Circuits". "3.3 Open-Ended Feedback" and "SLEQ Feedback" form the core of this part. Instead of "Results", this section can be titled such that it tells the reader now is time to read the feedback by the course participants. Section 4 "Discussion" should be combined in this section too.*

➢ The "Materials and Methods" section has now been replaced with the "Methodology" section of the paper (Section 2), and the outline restructured as recommended.
➢ A workflow diagram has been added as Figure 1 and the workflow diagram described with a summary at the beginning of Section 2.

*Outline: Thus, my new outline suggestion is:*

*1. Introduction*
*2. Methodology*
*Be more specific about stages of the class activity (i.e. pre-, during- and after).*
      *a. Course (activity) proposal and acceptance*
      *b. Technical setup material purchase*
      *c. Preparation of handouts, guidance material for students*
      *d. Implementation of the activity*

*e. FAQ*

*f. …*
*g. Design, Analysis and Collection of Written Feedback*
*3. Systems Approach and Hydrological Modelling*
*Try to create two or three subsections on the text provided in "2.1 Background". Be careful not to overlap with the Introduction part. In other words, avoid repetition.*
*4. Electronic Circuits*
*a. Theory and Construction*
*b. Classroom Application*
*It is too long as a whole, but some paragraphs are very short. Try to create sub-subsections to clearly present various aspects presented in Sec 2.3.*
*Manufacturing defects and choice of configuration shall be mentioned here.*
*c. Example Applications for Electronic Circuits*
*d. Modelling with Electronic Circuit Data*
*5. Feedback*
*a. Open-Ended Feedback*
*b. SLEQ Feedback*
*c. Feedback Loops (i.e. Sec 4 Discussion)*
*6. Conclusion*

➢ The manuscript outline has been re-structured as suggested.

*Conclusion: Overall not so good. I especially liked the paragraphs on P19 (L598-620). However, these are essentially the summary of feedback from the students. My suggestion is to relocate this into (new) Section 5 "Feedback". P18 L587-591 should be in the Methodology.*

➢ The paragraphs summarizing feedback from the students have been relocated into the Feedback section (pg. 16).
➢ P18 L587-591 in the first draft has been moved to the Methodology section (Section 2.5).

*Conclusion can be concise. It is more important to adopt a more general perspective into the scope and purpose of the study rather than sharing (again) various details of the activity. See also my comment on the Introduction. This (conclusion) is the part where the author can take the floor to convince the reader that hydrology education needs improvements which incorporate innovative class activities into teaching strategies adopted by the lecturers (and invite more lecturers to embrace this strategy). A discussion on the importance of teaching aspect of a "professorship" position can be given too. (e.g. a call for universities to accommodate such needs by distinguishing explicitly between "lecturer" and "researcher" positions. And that courses should be designed by a joint team so that well-defined teaching strategies supported with innovative and fitting class activities can be formulated addressing the needs of students and in line with emerging topics & technologies in hydrology.)*

➢ The conclusion has now been updated to show that hydrology education should be improved utilizing relevant classroom activities, as per these suggestions (Section 6, pg. 22)

*Specific comments*

*P2 L33. The connection to the previous paragraph can be strengthened in the following manner:*
*First, set the scene by mentioning that geography and environmental science teaching (recommended to) involve(s) different components, i.e.: introduction of conceptual theory, examples of how theories are implemented in real life applications and relevance to address societal challenges (addressed by the United Nations Sustainable Development Goals, UN SDGs), class activities to consolidate information to knowledge for achieving efficient learning of students. Then it is safe to bring forward the latter (class activities) and explain further. Paragraph 1 (in the current version of the manuscript) fits into here. Also, it will be good to include some comments on how such activities encourage development of a diverse set of skills, as well as if (by training future's professionals) and how such skills contribute to Multi-, Inter-, Trans- disciplinary research and practice.*

*Secondly, provide a brief overview on the progress and needs of hydrology education (see the list of suggested reading). Give some examples of class activities from the hydrology literature.*
*Thirdly, highlight the importance of data for science and teaching: The essential ingredient for geography and environmental sciences is data. Data collection is the backbone of advances in geosciences (thus teaching). In hydrological sciences education, water data (discharge and/or water level) collection is traditionally covered in many hydrology courses, however in rather a traditional way (e.g. in terms of measurement techniques) and often with a limited scope within a short duration. Maybe merge Paragraph 3 into this section. I recommend citing Tauro et al. (2017) to give an overview on the efforts to advance hydrological sciences through innovative measurement techniques. Also, the need for such efforts in view of not only providing support to financially disadvantaged countries but also harnessing the power of harmony in unity (e.g. WMO HydroHub).*

*Lastly, explain how electronic circuits are used for data collection purposes in hydrology and meteorology. Paragraph 2 fits into here, followed by Paragraph 4, 5 and 6.*

➢ The paper has been updated as per these suggestions regarding structure and content (Section 1).  Thank you for the detail presented in this specific comment and for the references.

*P2 L59-63. How is the open source electronics movement linked to the Open & FAIR data? Briefly mention.*

➢ The paragraph at the top of pg. 5 has been updated to mention the link between Open & FAIR data.  Please see the blue text.

*P3 L70. Before Paragraph 7, please state the research objective (and associated questions). L74-75 are the objectives of the course taught, not the manuscript's.*

➢ These objectives have now been added into this manuscript in the Introduction section of the manuscript (Section 1, bottom of pg. 5).

*P3 L76-93. Paragraph 8-9-10 should be moved to Section 2 "Materials and Methods" (Sec 2.3 in particular) as they describe several aspects related to pre-, during- and after the class activity (i.e. electronic circuit construction).*

➢ This text has now been moved to Section 2.5, edited and re-written for clarity.

P18 L587-591. Better if moved to Section 2 "Materials and Methods" (before subsections, a general summary can be given supported with a diagram showing all the stages).

  ➢ Please see Section 2.5 and Section 2.
  ➢ As suggested, a workflow diagram was added to Section 2 as Figure 1.

P19 (Code and Data Availability). Thanks for your dedication to Open Science and FAIR Data.

  ➢ Thanks for this comment.  Open Source and FAIR data are necessary for advancement of hydrology as a science.

P20 (Author contribution). Is there really a need to specify all tasks individually? Maybe one sentence is enough? E.g. All tasks (conceptualization, methodology, software, ….) are contributed by N. K.

  ➢ No, there is not a need to specify all tasks individually and the author contribution text has been updated as suggested.

P20 (Acknowledgements). Thanks for being so elaborate and honest.
P21 (References). Quite a comprehensive list, impressive. Properly cited, incorporated well into the text.

  ➢ Thank you for these comments.

*Minor Edits*

*P1 L14. "systems science, models in hydrology, and calibration" > "systems science, modelling in hydrology, and model calibration"*

  ➢ This line has now been updated in the Abstract.

*P3 L71. Abbreviation (USask) can be used for the "University of Saskatchewan" in the remaining part.*

  ➢ The abbreviation has been added throughout the text and the change is indicated in a blue color.

*P4 L115&122, P5 L160. Students in the Advanced Hydrology > Just say "students".*

  ➢ This has been updated; please see Section 3.1.

*P7 L213&L222, P12 L430, P17 L529… The name of the course "Advanced Hydrology class" doesn't need to be indicated every single time.*

  ➢ The course name has been removed throughout the document and reference is only made to the "class" rather than "Advanced Hydrology class."

*P13-15 L405-470. The texts quoted should be written in Italic.*

  ➢ All of the quoted texts are now written in italic and indicated in a blue color (Section 5.2).

*P18 L578. Add in parenthesis the English translation of "tawāw".*

> ➢ Due to restructuring of the Conclusion to not share again the details of the activity as suggested, this translation is now only provided on pg. 13.

*P18 L581. allowed for > allowed*

> ➢ Due to restructuring of the conclusion, the words "allowed for" have now been removed.

*P18 L584. Instead of "hydrology", use "technology".*

*P18 L586. Reformulate the sentence, maybe better as follows: "This paper provides explanations on Circuit theory and relevant concepts for classroom implementation and replication."*

> ➢ Due to restructuring of the paper and the removal of repeated information, these sentences have been removed.

[revised manuscript text omitted]

---

## Author Response (AR2)

**Author Response to Referees**

Thank you for these comments and recommendations. Responses are given below.

The referee response is given in *italic purple text*, whereas the author responses as a list of changes are given in non-italic text under each referee response. In the edited paper, changes to the text are indicated in a blue color.

**Referee #1**

*L53-76: This section could be shortened.*

Eleven lines at the beginning of the section has been removed to shorten this section (L53).

*L95-99: This is a long sentence. I recommend shortening it.*

P3: The sentence has been shortened and the revision is indicated by blue text.

*L155: The transition from introduction material to objectives is rather abrupt. Can the author articulate a research gap or need here that paves the way for the objectives?*

A research gap and need has been added to P5 before the objectives.

*Section 5.2: There are still a few instances in here where the author is providing student feedback on their teaching instead of the circuit activity only. Can these be removed?*

Student feedback provided on instructor teaching in Section 5.2 has been removed from the paper.

*Figure 12: I'm not sure this figure is helpful. I recommend removing it.*

Figure 12 has now been removed from the paper, along with the paragraph discussing this figure in Section 5.2.

**Referee #2: Nilay Dogulu**

*I would like to thank the author for improving the manuscript in line with the comments provided by the editor and reviewers. It is a long paper but it definitely conveys ideas and arguments clearly for the broader geosciences community. I am happy with the current version (only two suggestions >>> P3 L95: Dixon et al. 2020 shall be cited for WMO HydroHub too & P22 L688: teachers > lecturers). Thanks, Nilay.*

P3: Dixon et al 2020 has now been added to provide a citation near the WMO HydroHub link.

P21: The word "teachers" is now "lecturers" in the revised version.